# A Study of Vanadate Group Substitution into Nanosized Hydroxyapatite Doped with Eu^3+^ Ions as a Potential Tissue Replacement Material

**DOI:** 10.3390/nano12010077

**Published:** 2021-12-28

**Authors:** Nicole Nowak, Rafal Jakub Wiglusz

**Affiliations:** Institute of Low Temperature and Structure Research, Polish Academy of Sciences, Okolna 2, 50-422 Wroclaw, Poland

**Keywords:** nanosized hydroxyapatite, Eu^3+^ ion-doping, VO _4_^3^^−^ group substitution, photoluminescence, cytotoxicity, in vitro cell-culture study, tissue replacement material

## Abstract

In this study, nanosized vanadate-substituted hydroxyapatites doped with 1 mol% and 2 mol% Eu^3+^ ions were obtained via the precipitation method. To evaluate the structure and morphology of the obtained compounds, the XRPD (X-ray powder diffraction) technique, Rietveld refinement, SEM-EDS (scanning electron microscopy-energy-dispersive spectrometry) and TEM (transmission electron microscopy) techniques as well as FTIR (Fourier transform infrared) spectroscopy were performed. Moreover, the chemical formula was confirmed using the ICP-OES (Inductively coupled plasma optical emission spectroscopy spectroscopy). The calculated average grain size for powders was in the range of 25 to 90 nm. The luminescence properties of vanadium-substituted hydroxyapatite were evaluated by recording emission spectra and excitation spectra as well as luminescence kinetics. The crucial step of this research was the evaluation of the biocompatibility of the synthesized nanomaterials. Therefore, the obtained compounds were tested toward sheep red blood cells and normal human dermal fibroblast to confirm the nontoxicity and biocompatibility of new nanosized Eu^3+^ ion-doped vanadate-hydroxyapatite. Moreover, the final step of the research allowed us to determine the time dependent ion release to the simulated body fluid environment. The study confirmed cytocompatibility of vanadium hydroxyapatite doped with Eu^3+^ ions.

## 1. Introduction

The skeletal system of vertebrates is a very complex structure, not only due to the number and the variety of size or shape of the bones themselves, but also due to the complexity of tissues that are in the constant and inseparable neighborhood of bone [1,2].

The functional unit of bone is formed by concentric circles that surround a Haversian canal; the whole structure is called the osteon or Haversian system. This system creates space for nerves and blood vessels, thus enabling neurotransmission and nutrient delivery as well as the removal of metabolic products [2]. An equally important and inseparable part of bone structure is cartilage tissue, which adheres to bone structure and forms the articular surface. Bones also provide an attachment point for tendons, ligaments, and skeletal muscles and via the cooperation of all of these components, we can move our bodies around in a three-dimensional space [3].

Bone injuries, especially in the case of serious breakage such as open fractures are a problem not only in the regeneration of the bone tissue itself, but also in the tissues adjacent to the damaged bone such as cartilage, muscle, or nervous tissue as well as skin tissue [4,5,6]. In optimal conditions, repair processes can lead to the complete renewal of bone and soft tissue structure, however, the repair capacity of the nervous tissue is quite limited [7]. It is associated with a long recovery process and often does not bring the desired effects, especially if the damaged structures such as the skull and spine, protect the most important parts of the human body—the brain and spinal cord [8,9].

Therefore it is so important to choose the right strategy and treatment methods that will simultaneously stimulate the pool of stem cells within the body for faster regeneration of both the bones and the accompanying nerves [1,10,11].

After its success in the field of physical, chemical, and medical sciences, nanotechnology has now started revolutionizing the bio-detection and drug delivery sciences and bio regeneration techniques. The specific advantages include superior pharmacodynamics, pharmacokinetics, reduced toxicity, and targeting capability [12,13,14]. Biocompatibility and osteoinductive stimulation toward bone cells are well known properties of hydroxyapatite. Highly developed specific surface of synthetic hydroxyapatite generates the possibilities of forming a strong bond with living tissue such as bone and dental tissue [15,16]. Additionally, due to their photostability properties of rare-earth ion luminescence, they can be used as a biocompatible biosensor for cell or tissue imaging [17]. Both natural and synthetic hydroxyapatite is widely used to form three-dimensional scaffolds for bone and tooth filling [18,19,20]. It has been confirmed by many research studies that in particular, hydroxyapatite-based scaffolds when combined with biodegradable polymer biomaterials such as collagen, polylactide acid, chitosan, or alginate, produced promising effects due to their ability to act with living tissue, high biocompatibility, and induction of cell proliferation and growth of bone tissue [3,11,21,22,23,24].

As one of the rarest of the rare-earth elements and the most reactive element so far, europium exhibits relatively non-toxic effects on living cells [25,26,27]. Moreover, europium(III) ions can improve bone density and metabolism, especially when loaded in calcium-based compounds such as hydroxyapatite [28,29]. By obtaining the high crystal structure of the host material, Eu^3+^ ions loaded in hydroxyapatite show a high intensity of luminescence and can be used as a biocompatible biosensor for imaging cells or tissues. Chen et al. (2011) found another biomedical application for europium(III) ions using them as a drug carrier in ibuprofen-loaded Eu^3+^ ions: amorphous calcium phosphate (ACP) porous nanospheres. This composite showed biocompatibility toward porcine iliac artery endothelial cells (PIECs) and drug release experiments indicated sustained and slow release of the ibuprofen drug in simulated body fluid [30]. These properties of ibuprofen-loaded europium doped with hydroxyapatite were also investigated by Yang et al. (2008), who confirmed suitable drug release but, moreover, because of the presence of Eu^3+^ ions, the composite exhibited strong red luminescence and photoluminescence intensity, which increased with the increasing amount of the released drug. With its bioactive and luminescent properties, such a system presents the opportunity to easily monitor and track drug release, which offers prospective use in the field of drug delivery [31]. Additional confirmation of its use in the biomedical field is the fact that europium doped phosphors can be applied in medical uses as X-ray detectors in imagining systems [32,33].

Vanadium and vanadium compounds are commonly used in industry as a drying agent in paints, as a photographic developer, and is also used in black dyes, inks, or pigments for textiles, ceramics, and printing [34,35,36]. Due to its electron reduction potential, it is also used in vanadium redox flow batteries in electrochemical storage systems [37]. Surprisingly, vanadium, as a potentially toxic element, plays a quite significant role in living organisms. It is accumulated by tunicates and ascidians with the use of vanadium binding proteins in vacuoles of vanadocytes [38]. In organisms, vanadium is also present in fungal species such as Amanita in fungal fruiting bodies [39].

It has been experimentally proven to play a role in the neuroprotective and neuroregenerative pathway by inhibiting PTP—protein tyrosine phosphatase, which is responsible for the impairment of NGF (nerve growth factor) responsible for regeneration and the growth of neurons in the central nervous system of mammals. It has been shown that sodium orthovanadate inhibits PTP activity and therefore contributes to nerve growth in the rat hippocampus [40]. Moreover, in that connection, it was confirmed that sodium orthovanadate abolishes DNA breakage via inhibiting p53 and therefore arrests the process of apoptosis [41,42]. Another study confirmed that vanadium plays an essential role in the metabolism of rodents and determines proper physiological development in rats [43,44]. Regarding this study, it was also confirmed that local treatment with vanadate leads to strengthening of healing wounds by increasing cellular organization in the tissue structure [45,46].

The aim of this study was to obtain the europium doped hydroxyapatite materials substituted with vanadate groups (VO_4_^3^^−^) and evaluate their luminescence and biological properties primarily as a tissue filler in the wound healing process and their eventual use as a bioimaging material. For the first time, hydroxyapatite doped with Eu^3+^ ions and substituted with (VO_4_^3^^−^) groups were obtained. Structural study, luminescence properties as well as biocompatible features were evaluated, and the results clearly showed that two phased materials such as vanadate hydroxyapatites doped with Eu^3+^ ions are promising biocompounds for medical use as a skin tissue filler material. 

## 2. Materials and Methods

### 2.1. Synthesis Method

Two series of nanosized hydroxyapatite powders substituted with a vanadate group (VO_4_^3^^−^) in exchange for a orthophosphate group (PO_4_^3^^−^) and doped with Eu^3+^ ions with the chemical formula Ca_9.9_Eu_0.1_(PO_4_)_6−x_(VO_4_)_x_(OH)_2_ and Ca_9.8_Eu_0.2_(PO_4_)_6−x_(VO_4_)_x_(OH)_2_ (where x = 1, 2, 3, 4, 5, 6) were synthesized using the precipitation method. The substrates used in the synthesis were Ca(NO_3_)_2_∙4H_2_O (99.0–103.0% Alfa Aesar, Karlsruhe, Germany), Eu_2_O_3_ (99.99% Alfa Aesar, Karlsruhe, Germany), (NH4)_2_HPO_4_ (>99.0% Acros Organics (Thermo Fisher Scientific) Waltham, MA, USA), and NH_4_VO_3_ (≥99.0% Sigma-Aldrich, Saint Louis, MO, USA). 

The example synthesis for 2 g of Ca_9.9_Eu_0.1_(PO_4_)_5_(VO_4_)_1_(OH)_2_ nanopowder material, 4.5142 g of Ca(NO_3_)_2_∙4H_2_O was dissolved in 50 mL of distilled water and then 1.2748 g of (NH_4_)_2_HPO_4_ was dissolved separately in 50 mL of distilled water. To dissolve NH_4_VO_3_ in distilled water, the substrate (0.2258 g) was mixed together with 75 mL of distilled water and then placed in a Teflon vessel. The dissolving process was carried out in a microwave reactor (ERTEC MV 02-02) for 30 min at a temperature of 150 °C and under autogenous pressure (8–11 bar). The stoichiometric amount of Eu_2_O_3_ (0.0339 g) was digested in 0.2 mL HNO_3_ (≥65.0%, Sigma-Aldrich, Saint Louis, MO, USA) and 3 mL of distilled water to obtain water-soluble europium nitrate (Eu(NO_3_)_3_). The product was recrystallized three times to eliminate HNO_3_ residues by adding distilled water three times and evaporating at the temperature of 100 °C. Then, the obtained Eu(NO_3_)_3_ was dissolved in 25 mL distilled water and mixed with a water solution of Ca(NO_3_)_2_∙4H_2_O and then both substrates were mixed with a previously amalgamated water solution of (NH_4_)_2_HPO_4_ and water solution of NH_4_VO_3_. After rapidly amalgamating all substrates together, the pH was adjusted with 1.5 mL of ammonia (NH_3_∙H_2_O 25% Avantor, Gliwice, Poland) to achieve pH = 9. Synthesis was conducted using magnetic stirring (500 rpm) at a temperature of 150 °C. After the synthesis, the obtained composites were washed out in distilled water to obtain pH = 7 and were further dried for 2 days at 70 °C. Afterward, powders were thermally treated at a temperature of 600 °C for 6 h and the build-up temperature and cooling temperature was set up at 3 °C per minute. The syntheses of the remaining nanomaterials were analogous.

### 2.2. Physicochemical Characterization

#### 2.2.1. Material Characterization

With the use of the X-ray diffraction (XRD) technique, the vanadium-substituted hydroxyapatite doped with Eu^3+^ ion powders were examined to determine the crystalline structure of the obtained compounds. X-ray diffraction patterns were performed using a PANalytical X’Pert Pro X-ray diffractometer (Malvern Panalytical Ltd., Malvern, UK) with Ni-filtered Cu Kα radiation (U = 40 kV, I = 30 mA) in the 2θ range of 5–70°. The step time for XRD analysis was estimated with 0.05 and the time per step was estimated as 0.7 second per step. The XRD-recorded patterns were compared with the reference hydroxyapatite pattern from the Inorganic Crystal Structure Database (ICSD). The concentrations of Eu, Ca, V, and P in the resulting sample solutions were determined by the inductively coupled plasma-optical emission spectrometer (ICP OES) Agilent 720 (Santa Clara, CA, USA)(with standard setting). The samples were prepared by dissolving 100 mg of nanopowder material in 2 mL of 70% HNO_3_ (ASC, Sigma-Aldrich, Saint Louis, MO, USA) at the temperature 120 °C and by gradual adding of deionized water to the volume of 50 mL. The concentration of P, Eu, and V ions were measured in the solutions diluted 20 times and the concentration of Ca ions was measured in the solution diluted 500 times. For the measurements, three parallel samples of the solution were prepared and analyzed by the ICP-OES method (Agilent, model 720, Santa Clara, CA, USA), (with standard setting) and compared with standard curves in the concentration range of 0.05 to 5.00 mg/mL for Ca, Eu, and 100 to 200 mg/mL for P ions. To evaluate the presence of phosphate and vanadate groups in the structure of obtained compounds, IR spectra were measured in the range of 4000–400 cm^−1^ (mid-IR) at 295 K. The measurements of attenuated total reflectance ATR-FTIR were recorded with resolution 4 cm^−1^ (32 scans) using a Nicolet iS50 infrared spectrometer (Thermo Fisher Scientific, Waltham, MA, USA). The analysis of the elemental mapping of the selected sample was determined by using an FEI Nova NanoSEM 230 scanning electron microscope operating at an acceleration voltage in the range 3.0–15.0 kV and spot size of 4.0–4.5. The samples were prepared by evenly spraying a layer of graphite before observation. The morphology and nanostructure of the nanoparticles were investigated via high resolution transmission electron microscopy (HRTEM) using a Philips CM-20 Super Twin microscope operated at 200 kV. The selected material samples were prepared by the dispersion of powders in methanol. Then, a drop of suspension was deposited on a copper microscope grid covered with perforated carbon. 

#### 2.2.2. Luminescence Properties

The luminescence kinetics, emissions, and excitation spectra of vanadium-substituted apatite compounds doped with Eu^3+^ ions were determined with an FLS980 fluorescence spectrometer (Edinburgh Instruments, Kirkton Campus, UK). During the measurements of emission and excitation spectra, a 450W Xenon lamp was used as an excitation source and the radiation from the lamp was filtrated with a 300 mm monochromator equipped with a holographic grating (1800 grooves per mm, a blaze of 250 nm). To record the luminescence kinetics, a microsecond flashlamp (uF2) was used as a source of excitation and a Hamamatsu R928P photomultiplier (Hamamatsu, Hamamatsu City, Japan) was used as a detector. Both excitation and emission spectra were adjusted to the intensity of the excitation source according to the specifications of the device. The excitation spectra and luminescence kinetics were recorded at 618 nm according to the most intense electric dipole transition (from level ^5^D_0_ → ^7^F_2_ level) and excited at 396 nm [47,48,49].

### 2.3. Evaluation of Biocompatibility

#### 2.3.1. Preparation of Nanosized Vanadium-Substituted Hydroxyapatite Suspension

The stocks of nanosized vanadium-substituted hydroxyapatites doped with Eu^3+^ ions were prepared by the suspension of the used compounds in distilled water in the concentration of 1 mg/mL. Then, each stock was bath-sonicated for 1h at RT. Freshly prepared colloids were used in biological experiments.

#### 2.3.2. Cell Culture and Cytotoxicity Assay

Normal human dermal fibroblasts (NHDF, Sigma-Aldrich, Saint Louis, MO, USA) cell line was maintained in high glucose Dulbecco’s modified Eagle medium (DMEM) with L-glutamine (Biowest, Nuaillé, France) and supplemented with 200 U/mL penicillin and 200 µg/mL streptomycin and 10% heat-inactivated fetal bovine serum (FBS, South America origin, Biowest, Nuaillé, France). The cell line was incubated in standard conditions in a humified atmosphere of 95% air and 5% CO_2_ at 37 °C. The cell line was passaged three times before the experiments were performed. 

To evaluate their potential nontoxicity, the obtained compounds were tested on normal human dermal fibroblasts (NHDF) cell line (Sigma-Aldrich, Saint Louis, MO, USA) via the MTT cell viability assay. MTT, also known as the cytotoxicity assay, is a colorimetric assay used for establishing the percent of metabolically living cells. NHDF cells were seeded at a density of 10,000 cells per well in a 48-well plate and, after 24 h, when confluency was obtained, 60% to 70% cells were treated with selected compounds at two different concentrations of 50 µg/mL and 100 µg/mL. After 24 h of treatment with vanadium substitutes, hydroxyapatite composites doped with Eu^3+^ ions, the medium containing the tested compounds was removed and cells were washed out twice with sterile PBS (Biowest, Nuaillé, France) to remove detached and dead cells and to accurately rinse nanoparticles. After washing, freshly prepared MTT (Sigma-Aldrich, Saint Louis, MO, USA) reagent at a concentration of 0.5 mg per 1 milliliter was dissolved in sterile PBS and added to the cells that were treated with compounds and to the non-treated cells, which were used as a control group and were set up at cell viability of 100%. Cells were incubated for 3 h at 37 °C in a humified atmosphere of 95% air and 5% CO_2_. After the incubation process, PBS containing MTT was removed, and formazan crystals produced by metabolically active cells were dissolved by adding DMSO (Chempur, Piekary Śląskie, Poland). Absorbance was read at 560 nm and 670 nm (background reference). The experiment was conducted three times. The viability of the used cell lines was estimated using the following formula:$$Cell\;viability=\frac{sampleabsorbance}{controlabsorbance}\;\times 100\%$$

#### 2.3.3. Hemolysis Assay

Sterile and defibrinated sheep blood (Pro Animali) was washed out three times in sterile PBS and ultimately suspended in sterile PBS (Biowest, pH 7.4) at a ratio of 1:1. Selected vanadium-substituted hydroxyapatite nanoparticles were tested toward sheep red blood cells at concentrations of 50 µg/mL and 100 µg/mL. To establish positive control, sheep erythrocytes were combined with 10% SDS (sodium dodecyl sulfate) and treated as 100% of hemolysis, negative control was obtained by mixing sheep erythrocytes with sterile PBS. After 2 h of incubation at 37 °C, positive and negative control and red blood cell samples treated with selected hydroxyapatite-based composites were centrifuged (5000 RPM, 5 min) to obtain supernatant and the optical density was measured at 540 nm (Varioscan Lux). The hemolysis percentage was calculated using the formula below:$$Hemolysis=\frac{sampleabsorbance\;—\;negativecontrolabsorbance}{positivecontrolabsorbance\;—\;negativecontrolabsorbance}\times 100$$

Red blood cell morphology as well as the integrity of cell membrane were observed via confocal microscopy (Olympus IX83 Fluoview FV 1200, 10× magnification with additional 4× digital magnification). Sheep erythrocytes were prepared as described above and treated with selected compounds; positive and negative controls were also prepared. After 2 h of incubation at 37 °C, red blood cells were centrifuged (5000 RPM, 5 min), the supernatant was gently removed, and cell precipitate was suspended with sterile PBS at a ratio of 1:1. A blood smear was prepared by transferring 5 µL of the sample onto a microscope slide and using a coverslip to obtain the smear.

#### 2.3.4. Time Dependent Ion Release to SBF

Simulated body fluid (SBF) solution with an optimal value of pH and physiological temperature closely mimics blood plasma in the human body. By using SBF, the rate of ion release from the tested biomaterials to the fluid environment can be easily evaluated, especially when further in vitro and in vivo tests are planned [50,51]. To evaluate ion release, the two representatives from the two obtained nanopowder series materials were selected. Ca_9.9_Eu_0,1_(PO_4_)_5_(VO_4_)_1_(OH)_2_ and Ca_9.8_Eu_0,2_(PO_4_)_5_(VO_4_)_1_(OH)_2_ nanopowder materials were used in this experiment because the XRD diffractograms showed a clear hexagonal structure of hydroxyapatite. These were placed in the Falcon tubes separately and the previously prepared simulated body fluid (pH = 7.40) was gently added to obtain a final concentration of 1 mg/mL. The simulated body fluid was prepared by accurately following the procedure created by Kokubo et al. [50]. The samples were placed in the shaker incubator and the temperature was set to 37 °C with a rotation of 100 rpm. The period when samples were collected was set to 0 min, 5 min, 15 min, 30 min, 45 min, 60 min, 360 min, and 1440 min of incubation with simultaneous rotation. Each time, 3 mL of the fluid sample was collected in a new, separated Falcon tube, and 3 mL of fresh SBF was added to the remaining solution to refill the missing volume. Subsequently, when all fluid samples were collected, 0.2 mL of 70% HNO_3_ (ASC, Sigma-Aldrich) was added to all representatives and deionized water was added to obtain a final volume of 25 mL. When all samples were prepared, the presence of investigated ions such as Ca, P, Eu and V was identified by the inductively coupled plasma-optical emission spectrometer (ICP OES) Agilent 720 instrument.

## 3. Results and Discussion

### 3.1. Analysis of Structure and Morphology

Two series of hydroxyapatite-based nanopowders doped with 1 mol% Eu^3+^ ions and 2 mol% Eu^3+^ ions and substituted with different amounts of vanadate groups were synthesized using the precipitation method. XRD diffractograms clearly showed the hexagonal structure of hydroxyapatite for powder materials that contained 1 mol% Eu^3+^ ions and 2 mol% Eu^3+^ ions and substituted with up to two vanadate groups substituted in the place of phosphate groups (Figure 1a,b). The delicate signals of hydroxyapatite hexagonal structure can be noticed among samples that contain 1 mol% Eu^3+^ and 2 mol% Eu^3+^ and are substituted with three (VO_4_^3^^−^), however, the more vanadium appeared in the sample, the more the calcium pyrovanadate phase was visible. The results for two series of nanosized materials containing up to two vanadate groups, corresponded to the standard ICSD database diffractogram pattern for hydroxyapatite crystals (ICSD-262004). For the above-mentioned powders, signals in the range of 32° to 34° in the experimental patterns corresponded to distinctive phosphate groups of the hydroxyapatite crystal structure (ICDS-262004). Broader bases of the peaks, especially in the range of 32° to 34°, may indicate the nanosized structure of the nanosized materials substituted with (VO_4_^3^^−^) 1–3 and doped with 1 mol% and 2 mol% of Eu^3+^ ions (Figure 1a,b) [52,53]. The gradual increase in the number of vanadate groups in samples of the obtained nanopowder materials eventually led to the gradual decrease in the intensity of the signal from the phosphate groups and the increase in the intensity of the signal from the vanadate groups, which is certainly observed in the XRD diffractograms (Figure 1a,b). Interestingly, the occurrence of another phase was noticeable in the range of 27°–32° for the XRD experimental patterns of both series of nanopowder materials that were substituted with three and more vanadate groups. It appears that a progressive increase in the intensity of the signal from the range 27°–32° came from Ca_2_V_2_O_7_ (calcium pyrovanadate) (Figure 1a,b). Experimental data of the obtained materials substituted with (VO_4_^3^^−^) 3–6 groups were compared with the ICSD XRD pattern of Ca_2_V_2_O_7_ (ICSD-421266). Our results indicate that the precipitation method used in the experiment was sufficient to obtain materials substituted with up to two (VO_4_^3^^−^) groups in place of (PO_4_^3^^−^) groups in the crystal hydroxyapatite structure. The hydrothermal synthesis method seems to be more adequate to incorporate more than three vanadate groups into the hydroxyapatite crystalline framework [54]. Nonetheless, there were also some data that indicate that the ammonia environment is not suitable to obtain vanadate-substituted hydroxyapatite. To obtain an alkaline environment of a chemical reaction during the synthesis of vanadate-substituted hydroxyapatite, NaOH should be substituted for NH_3_∙H_2_O [54,55]. Moreover, other substrates can be used during the synthesis of vanadate-substituted hydroxyapatite. Some data suggest that V_2_O_5_ could be used as a substitute of NH_4_VO_3_ and P_2_O_5_ as a substitute of (NH_4_)_2_HPO_4_ [54,55,56,57].

The structural refinement was calculated by the Maud program (version 2.99) and was based on the hexagonal structure of hydroxyapatite and triclinic calcium pyrovanadate crystals indexing of the CIF (Crystallographic Information File) [58,59]. The quality of the structural refinement was evaluated via R-values (see Appendix A and Appendix A). Presence of apatite structure as well as secondary phase formation of calcium pyrovanadate among the nanopowder materials Ca_9.9_Eu_0.1_(PO_4_)_6−x_(VO_4_)_x_(OH)_2_, and Ca_9.8_Eu_0.2_(PO_4_)_6−x_(VO_4_)_x_(OH)_2_ (where x is equal 1, 2, 3, 4, 5 and 6) was confirmed. Moreover, the calculated average grain size for powders was in the range of 25 to 90 nm. More details regarding Rietveld refinement are presented in the Appendix A. 

The FTIR spectra of the second series of nanopowder materials containing Ca_9.8_Eu_0.2_(PO_4_)_6−x_(VO_4_)_x_(OH)_2_(x = 1, 2, 3, 4, 5 and 6) confirmed the crystalline hydroxyapatite structure due to the presence of characteristic active vibrational bands that refer to hydroxyl groups (OH^−^) and most importantly to phosphate groups (PO_4_^3^^−^) (Figure 2). The absorption bands of the phosphate group at 560.70 cm^−1^ and 600.24 cm^−1^ corresponded to the double degenerate bending mode (ν2) of the P–O–P bonds and triply degenerate bending mode (ν4) of the P–O bonds, respectively [18,60]. The absorption bands of the phosphate group at 962.34 cm^−1^ and 1086.20 cm^−1^ correlated with the non-degenerative symmetric stretching mode (ν_1_) of P–O and the triply degenerative asymmetric stretching mode (ν_3_) of the P–O bond, respectively [60,61,62]. All positions of the bands corresponded exactly to the hydroxyapatite structure, but only in compounds that contain up to three (VO_4_^3^^−^) groups. It is also noticeable (Figure 2) that the additional incorporation of (VO_4_^3^^−^) groups into the hydroxyapatite framework unalterably entails the shift in the absorption bands toward lower wavelengths [63]. Gradual increment of the number of vanadium groups substituted for phosphate groups leads to the appearance of characteristic vibrational bands that invoke the appearance of Ca_2_V_2_O_7_ crystal structure [63]. It is particularly observed for the sample with the highest number (x = 6) of vanadate groups substituted for phosphate groups. Typical vibrational bands at 417.03 cm^−1^ (ν_4_) and 561.66 cm^−1^ (ν_3_) refer to the asymmetric vibration models of O–Ca–O and O–V–O bending, respectively. The vibrational bands that are noticeable at 825.38 cm^−1^ (ν_1_) and 760.78 cm^−1^ (ν_2_) corresponded to the stretching frequencies of the V–O group [64,65]. Additionally, a narrow vibrational band at 3567.18 cm^−1^ was seen due to stretching frequencies of the surface-absorbed water [64,65,66]. The described characteristic bonding of O–Ca–O, O–V–O, and V–O groups was the most visible for the sample material with the greatest number of vanadium groups (VO_4_^3^^−^). Fewer vanadium groups incorporated into the hydroxyapatite structure led to the smaller band vibrations occurring; additionally, the characteristic shift into the direction of hydroxyapatite was visible and typical phosphate groups (PO_4_^3^^−^) could be noticed [61,62].

The TEM images clearly show the nanostructure of Ca_9.8_Eu_0.2_(PO_4_)_5_(VO_4_)_1_(OH)_2_, Ca_9.8_Eu_0.2_(PO_4_)_4_(VO_4_)_2_(OH)_2_, and Ca_9.8_Eu_0.2_(PO_4_)_3_(VO_4_)_3_(OH) powders (Figure 3). Additionally, according to the SAED (selected area electron diffraction) technique, all selected materials presented well developed spotty rings, which signify the crystalline structure of the obtained powders (Figure 3) [67,68]. As also observed on the images (Figure 3), nanosized particles had the tendency to form into larger agglomerates. The results of ICP-OES measurements showed the presence and concentration of Eu^3+^ ions in the compounds containing 2 mol% of the doped lanthanide in the samples of Ca_9.8_Eu_0.2_(PO_4_)_6−x_(VO_4_)_x_(OH)_2_ where “x” is equal 1, 2, and 3. The presence of the vanadium element was also confirmed as well as calcium and phosphorous (Table 1) and the content of the elements approximately matched the theoretical number of particular elements in the obtained nanosized materials. Indeed, all desired ions were present in the nanomaterials, and the ICP-OES measurements showed an almost identical content of elements when compared with theoretical calculations. However, the content of vanadium ions seemed to be less when compared to the formula that was previously established. These data resulted from the formula for the hydroxyapatite crystalline calculation, hence it can be observed that phosphorus is indeed substituted for vanadium, but in a lower amount than assumed. Simultaneously, the second phase of calcium pyrovanadate appeared (Figure 1 and Figure 2) and was probably the result of more vanadium ions being incorporated into this structure and not into the hydroxyapatite lattice. Moreover, the SEM-EDS mapping of Ca_9.8_Eu_0.2_(PO_4_)_4_(VO_4_)_2_(OH)_2_ confirmed the presence of all theoretical components such as oxygen, phosphorous, vanadium, and europium in the hydroxyapatite crystalline structure. The performed analysis also showed that all the components were equally distributed over the entire surface of the tested material (Figure 4).

### 3.2. Luminescence Properties

Based on the XRD results (Figure 1) and partially on the FTIR spectra results (Figure 2), it is clear that in both series of obtained compounds, only those that contained 1 mol% and 2 mol% of europium (III) ions and were substituted with one and two (VO_4_^3^^−^) groups characterized by crystalline hydroxyapatite structure. The more phosphate groups is substituted by vanadate groups the less of pure hydroxyapatite structure is visible and more of calcium pyrovanadate appears. Taking into consideration the duality of the obtained compounds, we wanted to evaluate whether the different crystalline phases influenced the luminescence properties of Eu^+3^ ions and whether they decreased or increased these features. Therefore, the presence of Eu^3+^ ions incorporated into the structure of materials based on hydroxyapatite was confirmed by performing luminescence studies. The good quality emission spectra of both series of vanadium hydroxyapatite compounds Ca_9.9_Eu_0.1_(PO_4_)_6−x_(VO_4_)_x_(OH)_2_ and Ca_9.8_Eu_0.2_(PO_4_)_6−x_(VO_4_)_x_(OH)_2_ ( x= 1, 2, 3, 4, 5 and 6) were measured in the spectral range of 500 to 750 nm (Figure 5a,b). During the measurements of both series of materials, an excitation wavelength of 396 nm was set as a function of the concentration of optically active ions. The recorded spectra were normalized to the characteristic europium transition ^5^D_0_ → ^7^F_1_. Five typical transitions of Eu^3+^ ions were present in the spectra at wavelengths of 575 nm, 585 nm, 618 nm, 660 nm, and 710 nm, which corresponded to the transition from the excited level of ^5^D_0_ to the levels of ^7^F_0-4_. The transitions were assigned as ^5^D_0_ → ^7^F_0_, ^5^D_0_ → ^7^F_1_, ^5^D_0_ → ^7^F_2_, ^5^D_0_ → ^7^F_3_, and ^5^D_0_ → ^7^F_4_, respectively, with increasing wavelength value. The most intense peak corresponded to the ^5^D_0_ → ^7^F_2_ transition, for which emission was observed at wavelengths in the range of 600–625 nm, while the maximum intensity was observed at 618 nm (see Figure 5a,b) [47,60,69]. A clear red emission from Eu^3+^ ions incorporated into vanadate hydroxyapatite materials was observed. According to the Judd–Ofelt theory, the ^5^D_0_ → ^7^F_0_ transition is strictly forbidden and its occurrence indicates a violation of the selection rules of the above-mentioned theory [47]. By analyzing the canonical transition ^5^D_0_ → ^7^F_0_, the number of crystallographic sites substituted by europium ions into the structure of the host material can be assumed. The appearance of this transition indicates the location of Eu^3+^ ions at the low-symmetry environment and its observation is enabled when Eu^3+^ ions occupy sites with local symmetry of C_n_, C_nv_, C_s_ [70]. It can be seen that in the case of transition from the level ^5^D_0_ to ^7^F_0_ for the materials that contained only one (VO_4_^3^^−^) group substituted for the (PO_4_^3^^−^) group, the band clearly stood out from the spectra of other compounds. The transition ^5^D_0_ → ^7^F_0_ of materials Ca_9.9_Eu_0.1_(PO_4_)_6−x_(VO_4_)_x_(OH)_2_; Ca_9.8_Eu_0.2_(PO_4_)_6−x_(VO_4_)_x_(OH)_2_ (x = 1) was noticeably divided into three splits, which indicates the occupancy of three different crystallographic sites with the local symmetry of C_n_, C_nv_, C_s_ in the hydroxyapatite structure by Eu^3+^ ions [71]. As the ^7^F_0_ level is degenerate, it does exhibit crystal field splitting and our results showed at least three different emitting species [48]. In the case of further transition from the level ^5^D_0_ to ^7^F_1_, the same tendency was observed that for the above-mentioned materials with only one substituted (VO_4_^3^^−^) group, the band was broad and not visibly split as it was in the other materials that contained two and more (VO_4_^3^^−^) groups. For compounds Ca_9.9_Eu_0.1_(PO_4_)_6−x_(VO_4_)_x_(OH)_2_; Ca_9.8_Eu_0.2_(PO_4_)_6−x_(VO_4_)_x_(OH)_2_ where x < 2, 3, 4, 5, 6, the transition ^5^D_0_ → ^7^F_1_ is divided into three splits and these results correspond to other studies, where such characteristic splits occurred [72,73]. The most intensive transition for both series of the tested nanomaterials appeared to be the so-called hypertensive transition from level ^5^D_0_ to ^7^F_2_. For this transition, the same trend was maintained as for compounds Ca_9.9_Eu_0.1_(PO_4_)_5_(VO_4_)_1_(OH)_2_ and Ca_9.8_Eu_0.2_(PO_4_)_5_(VO_4_)_1_(OH)_2_, where one broad band was visible. The hypertensive transition for the rest of the compounds was divided into two distinct bands, which correspond to the results obtained by other studies [63,72].

The excitation spectra were measured in the wavelength range of 240–600 nm for which the emission was monitored at 618 nm (see Figure 6). The recorded spectra were normalized to the characteristic europium transition ^7^F_0_ → ^5^D_2_. The spectra for both series of compounds Ca_9.9_Eu_0.1_(PO_4_)_6−x_(VO_4_)_x_(OH)_2_ and Ca_9.8_Eu_0.2_(PO_4_)_6−x_(VO_4_)_x_(OH)_2_ (x = 1, 2, 3, 4, 5, 6) showed peaks from the transitions ^7^F_0_ → ^5^H_(3–7)_, ^7^F_0_ → ^5^L_6_, and ^7^F_0_ → ^5^D_2_. In the spectra recorded for Ca_9.8_Eu_0.2_(PO_4_)_6−x_(VO_4_)_x_(OH)_2_, where x ranged from 1 to 6, peaks corresponding to the transitions ^7^F_0_ → ^5^F_(1–4)_ and ^7^F_0_ → ^3^P_0_ could also be seen. In the (VO_4_^3^^−^)_x_ spectra, where x ranged from 1 to 5, apart from the transition’s characteristic for Eu^3+^ ions, an intense peak was visible at a wavelength of approximately 270 nm, corresponding to the charge transfer of an electron between the ionized oxygen atom and the europium ion (O^2^^−^→Eu^3+^). Our results are compatible with other results that provided data of charge transfer between O^2^^−^ and Eu^3+^ ions in the hydroxyapatite structure [60,70,74]. For (VO_4_^3^^−^)_6_ spectra, this peak was masked with a more intense peak, for which the maximum of intensity was visible at a wavelength lower than 240 nm. With the increasing number of vanadate groups in the hydroxyapatite structure, the intensity of the broad signal originating from the electron transfer from the oxygen ion O^2^^−^ to the vanadium ion, V^5+^, (O^2^^−^ → V^5+^) increases. This is a natural tendency, and it is natural for the charge transfer to be increased with an increased number of vanadium groups substituted for phosphate groups. Similar results have been presented in different studies, and it seems to strongly correspond to our results [63,72,73,75,76]. The maximum intensity of this peak was noticeable on the spectra corresponding to the samples of Ca_9.9_Eu_0.1_(VO_4_)_6_(OH)_2_ and Ca_9.8_Eu_0.2_(VO_4_)_6_(OH)_2_ and appeared at wavelengths in the range 380–400 nm. It is clearly visible for both series of obtained samples as for the compounds: Ca_9.9_Eu_0.1_(PO_4_)_6−x_(VO_4_)_x_(OH)_2_ and Ca_9.8_Eu_0.2_(PO_4_)_6−x_(VO_4_)_x_(OH)_2_ (x = 1); the charge transfer O^2^^−^ → Eu^3+^ was the most visible and clear characteristic of excitation spectra for europium ions. The more vanadate groups appeared, the more intense the charge transfer O^2^^−^ → V^5+^ appeared and the results referred to both series of materials [72,73,75]. Moreover, the charge transfer from oxygen to vanadium seems to be slightly shifted toward the highest wavelength number, however, it can be caused by the incorporation of the vanadium groups into the hydroxyapatite framework. Nevertheless, the peak positions for the excitation and emission spectra were in agreement with those expected for Eu^3+^ ions incorporated in exchange for calcium(II) ions into the hydroxyapatite structure [30,31,47,74].

For both 1 mol% and 2 mol% Eu^3+^ ion-doped hydroxyapatites, there was a significant difference in the decay time when more than one vanadate group was substituted instead of the phosphate group. For samples recorded for 1 mol% Eu^3+^:CaHAp with (VO_4_^3^^−^)_x_ groups and 2 mol% Eu^3+^:CaHAp with (VO_4_^3^^−^)_x_ groups where x ranges from 2 to 6, the decay time was similar, much shorter than for hydroxyapatites containing only one vanadate group (see Figure 7). Such a tendency corresponds to the emission and excitation spectra recorded for both series of compounds (Figure 5 and Figure 6) where the difference can be observed between compounds containing only one vanadium group in the hydroxyapatite structure and the rest of the compounds with a higher number of (VO_4_^3^^−^) groups.

### 3.3. Biological Properties

It has been decided that the biocompatibility of selected compounds from both series would be evaluated. Therefore, from the first series, which contained a constant concentration of Eu^3+^ ions (1 mol%), compounds with 1, 2, 3, and 6 vanadate groups were selected. From the second series with a 2% concentration of europium(III) ions, the compounds were selected analogously to the first series. The compounds with 1–3 vanadate groups substituted with phosphate groups were selected for the evaluation of potential toxicity because it was noted that our compounds maintained the hexagonal structure of hydroxyapatite only up to three substituted groups. Moreover, compounds that contained the greatest number of vanadate groups were chosen for this experiment to establish whether the increased concentration of vanadium elements showed a potential toxic effect. It was found that the hemolysis assay (Figure 8) showed nontoxicity of the tested compounds and hemoglobin release was estimated below 5% of acceptable hemolysis, which naturally occurs in the blood system [77]. The results were compared to red blood cells treated with 1% SDS (dodecyl sulfoxide), which was established as the positive control and caused 100% of hemoglobin release and cell membrane disruption (data not shown). The negative control, which was maintained at ~1% of hemoglobin release, was obtained by treating purified erythrocytes with PBS buffer (pH = 7.4). Data revealed that all of the compounds showed biocompatibility. Moreover, the shape of red blood cells remained round, and no pathological alterations of the cell membrane were observed (Figure 9).

The results clearly showed that the selected compounds showed biocompatible properties toward the NHDF cell line (normal human dermal fibroblasts). It was decided that marginal compounds of the two series would be tested because of the obtained hemolysis assay results. The viability of cells was maintained at around 100 percent even when cells were treated with the highest concentration of prepared double distilled water-based colloids (Figure 10). The safest concentration of the tested compounds was 50 µg/mL where the viability of the NHDF cell line was slightly more increased when compared to cells treated with a concentration of 100 µg/mL. While many studies indicate the toxicity of the vanadium compounds such as ammonium metavanadate, calcium orthovanadate, and calcium pyrovanadate toward mammalian gastrointestinal, respiratory, urinary, and reproductive systems, our compounds showed non-toxic properties toward red blood cells and normal human dermal fibroblasts [78,79,80]. The toxicity of the vanadium element is mainly caused by the overdosage of this component, as has been evaluated in some research [78]. However, on the other hand, there is much evidence that indicates the positive influence on living organisms, for example, neuroprotective and neuroregenerative properties [41,42,44]. Nevertheless, neither the high concentration used in the case of our materials (100 µg/mL) nor the increased number of vanadium groups incorporated into the hydroxyapatite structure caused a harmful effect on erythrocytes and the NHDF cell line.

### 3.4. Ion Release to SBF

The ion dependent release to the simulated body fluid environment was evaluated from two selected nanopowder materials Ca_9.9_Eu_0,1_(PO_4_)_5_(VO_4_)_1_(OH)_2_ and Ca_9.9_Eu_0,2_(PO_4_)_5_(VO_4_)_1_(OH)_2_ (Table 2 and Table 3). Ca_9.9_Eu_0,1_(PO_4_)_5_(VO_4_)_1_(OH)_2_ and Ca_9.8_Eu_0,2_(PO_4_)_5_(VO_4_)_1_(OH)_2_ nanopowder materials were used in this experiment because the XRD diffractograms showed a clear hexagonal structure of hydroxyapatite (Figure 1). Therefore, during the experiment, the ion release from the solid hydroxyapatite crystalline materials was determined. The results showed that Ca and P ions were already present in the fluid in both samples in the concentration 11 ± 0.5 ppm for calcium and 2.45 ± 0. ppm to 2.84 ± 0.15 ppm for phosphorus ions (Table 2 and Table 3). Their appearance comes from the components of simulated body fluid itself [50,51]. Nevertheless, a gradual increase was noticed during longer incubation and after 1440 h, calcium concentration was maintained at 13.84 ± 0.7 ppm for Ca_9.9_Eu_0,1_(PO_4_)_5_(VO_4_)_1_(OH)_2_ to 14.9 ± 0.7 ppm Ca_9.9_Eu_0,2_(PO_4_)_5_(VO_4_)_1_(OH)_2_, and for phosphorus ions, it was maintained around 2.8 ± 0.15 ppm for Ca_9.9_Eu_0,1_(PO_4_)_5_(VO_4_)_1_(OH)_2_ and 2.9 ± 0.15 ppm for Ca_9.9_Eu_0,2_(PO_4_)_5_(VO_4_)_1_(OH)_2_ nanopowder materials (Table 2 and Table 3). Slightly different observations were seen for vanadium ion release because for Ca_9.9_Eu_0,1_(PO_4_)_5_(VO_4_)_1_(OH)_2_, vanadium ion release was already observed after 45 min of incubation and for Ca_9.8_Eu_0,2_(PO_4_)_5_(VO_4_)_1_(OH)_2_ after 30 min and also had a tendency for gradual release (Table 2 and Table 3). There was no release observed for Eu ions, however, slight detection was observed for the sample doped with higher lanthanide concentration (Table 2 and Table 3). For the two tested representatives, all investigated ions appeared to have a relatively slow release to the simulated body fluid, which is a promising result, and they can be used in our future in vitro and in vivo tests.

## 4. Conclusions

This paper presents the structural characterization, luminescence, and biological properties of vanadium hydroxyapatite doped with 1 mol% and 2 mol% of Eu^3+^ ions. The samples obtained via the precipitation method and thermally treated at 600 °C showed a hydroxyapatite hexagonal structure up to three vanadate groups substituted for phosphate groups. It was confirmed by X-ray diffractometry and FTIR spectra that the gradual increase in (VO_4_^3+^) groups in the obtained nanopowder materials eventually led to the gradual decrease in the intensity of the signal from (PO_4_^3+^) groups of the hydroxyapatite framework and an increase in the intensity of the signal from (VO_4_^3^^−^) groups of calcium vanadate. The luminescence study showed the characteristic red emission spectra of Eu^3+^ ions doped among all samples. Our study also presented how the number of vanadate groups in europium-doped hydroxyapatite influences the emission spectra, excitation spectra, and luminescence kinetics. Finally, the evaluation of the potential toxicity of the obtained nanomaterials confirmed hemocompatibility toward sheep red blood cells even in the highest tested concentration. Furthermore, our study confirmed cytocompatibility of vanadium hydroxyapatite doped with Eu^3+^ ions and our materials exhibited biocompatibility even when the highest number of vanadium groups was incorporated into hydroxyapatite. The time dependent ion release experiment showed slow and gradual element release to the SBF solution and additionally confirmed the potential biological application of the obtained nanopowder materials.

## Figures and Tables

**Figure 1 nanomaterials-12-00077-f001:**
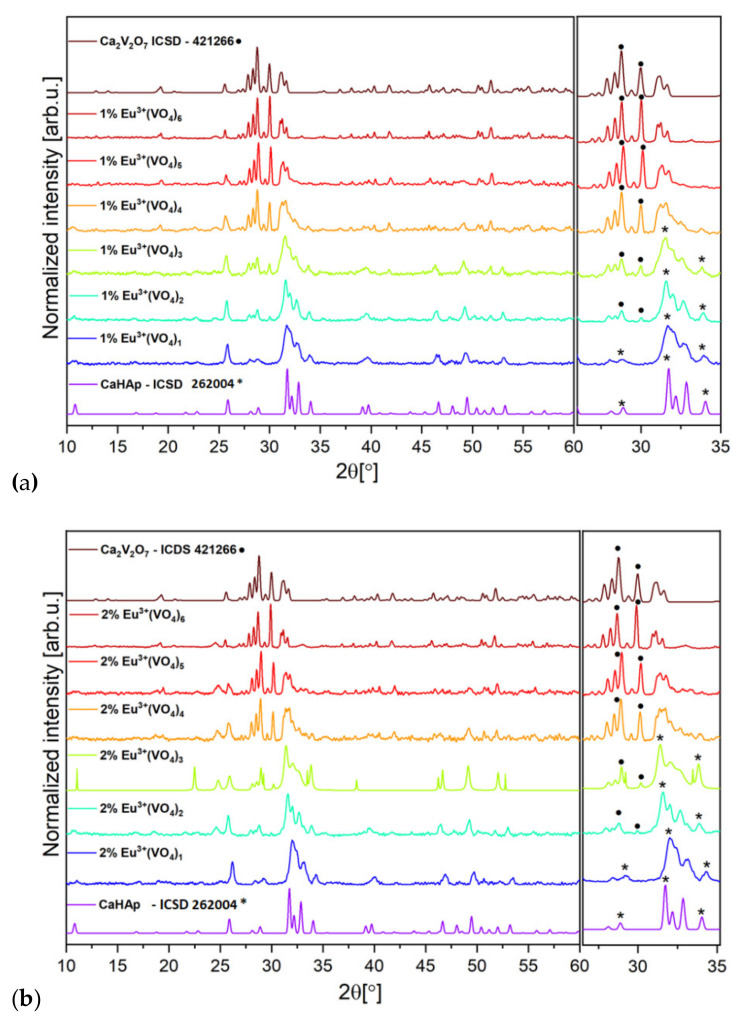
XRD results of (**a**) Ca_9.9_Eu_0.1_(PO_4_)_6−x_(VO_4_)_x_(OH)_2_, and (**b**) Ca_9.8_Eu_0.2_(PO_4_)_6−x_(VO_4_)_x_(OH)_2_, where x is equal 1, 2, 3, 4, 5, and 6. The obtained materials were thermally treated at a temperature of 600 °C for 6 h. The XRD results were compared with the ICSD database hydroxyapatite and calcium pyrovanadate patterns. The signals from hydroxyapatite are labeled with (*****) and signals from calcium pyrovanadate are labeled with (•).

**Figure 2 nanomaterials-12-00077-f002:**
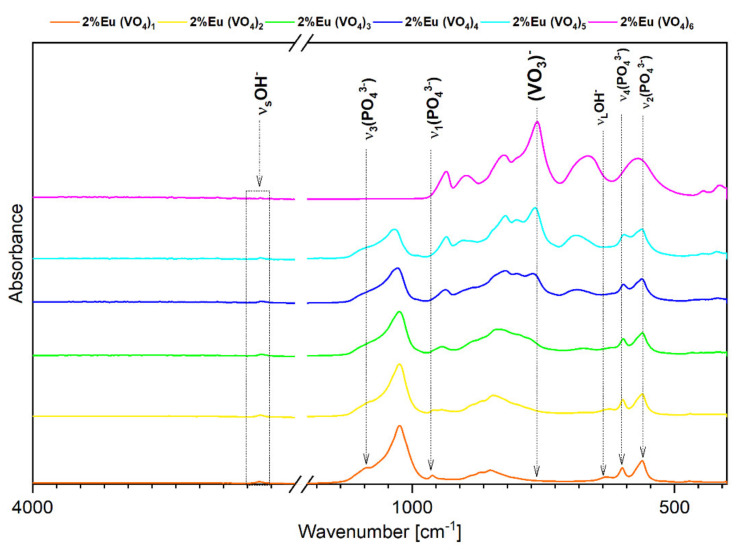
FTIR spectra of Ca_9.8_Eu_0.2_(PO_4_)_6−x_(VO_4_)_x_(OH)_2_, where x = 1, 2, 3, 4, 5, and 6.

**Figure 3 nanomaterials-12-00077-f003:**
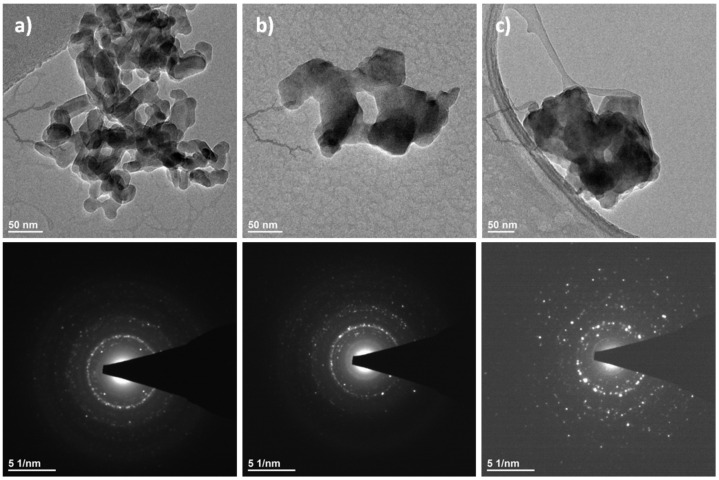
TEM and SAED images of (**a**) Ca_9.8_Eu_0.2_(PO_4_)_5_(VO_4_)_1_(OH)_2_, (**b**) Ca_9.8_Eu_0.2_(PO_4_)_4_(VO_4_)_2_(OH)_2_, and (**c**) Ca_9.8_Eu_0.2_(PO_4_)_3_(VO_4_)_3_(OH)_2_.

**Figure 4 nanomaterials-12-00077-f004:**
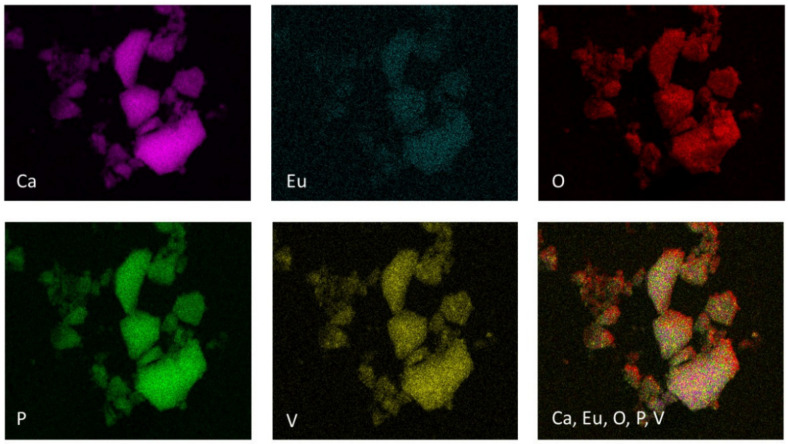
SEM-EDS mapping of Ca_9.8_Eu_0.2_(PO_4_)_4_(VO_4_)_2_(OH)_2_ compound showing the elemental composition: calcium, phosphorous, europium, vanadium, and oxygen.

**Figure 5 nanomaterials-12-00077-f005:**
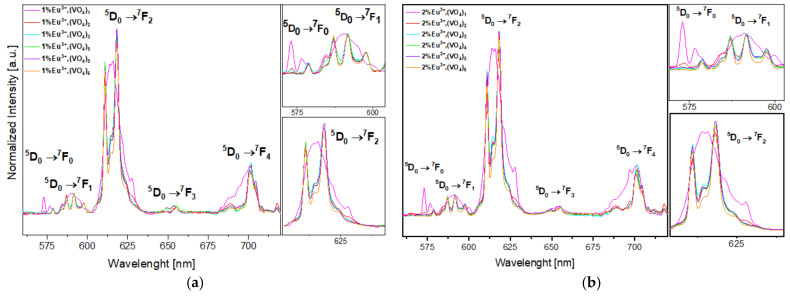
Emission spectra of (**a**) Ca_9.9_Eu_0.1_(PO_4_)_6−x_(VO_4_)_x_(OH)_2_, and (**b**) Ca_9.8_Eu_0.2_(PO_4_)_6−x_(VO_4_)_x_(OH)_2_, where x is equal 1, 2, 3, 4, 5, and 6.The excitation wavelength is equal to 396 nm, a 525 nm filter was used in this experiment.

**Figure 6 nanomaterials-12-00077-f006:**
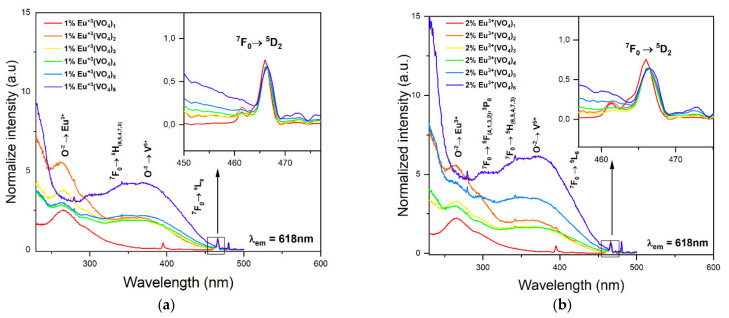
Emission spectra of (**a**) Ca_9.9_Eu_0.1_(PO_4_)_6−x_(VO_4_)_x_(OH)_2_ and (**b**) Ca_9.8_Eu_0.2_(PO_4_)_6−x_(VO_4_)_x_(OH)_2_, where x is equal 1, 2, 3, 4, 5, and 6.The excitation wavelength was equal to 396 nm and a 525 nm filter was used in this experiment.

**Figure 7 nanomaterials-12-00077-f007:**
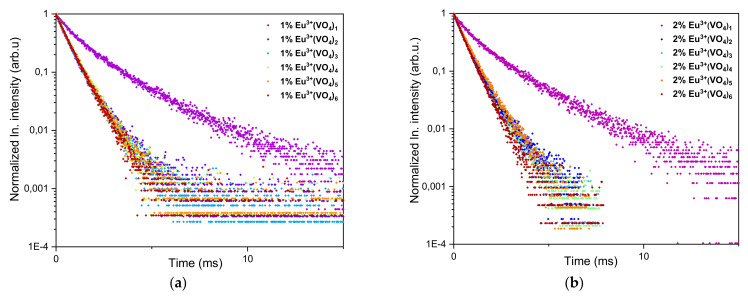
Emission spectra of (**a**) Ca_9.9_Eu_0.1_(PO_4_)_6−x_(VO_4_)_x_(OH)_2_ and (**b**) Ca_9.8_Eu_0.2_(PO_4_)_6−x_(VO_4_)_x_(OH)_2_, where x is equal 1, 2, 3, 4, 5, and 6.The excitation wavelength was equal to 396 nm and a 525 nm filter was used in this experiment.

**Figure 8 nanomaterials-12-00077-f008:**
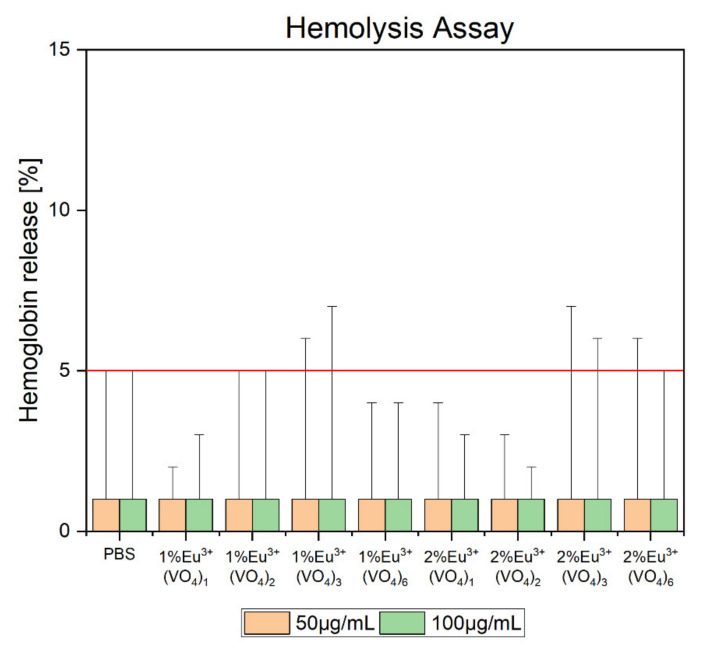
Hemolysis of selected compounds from the first series: Ca_9.9_Eu_0.1_(PO_4_)_6−x_(VO_4_)_x_(OH)_2_, and second series: Ca_9.8_Eu_0.2_(PO_4_)_6−x_(VO_4_)_x_(OH)_2_, where x = 1, 2, 3, 6. The concentration of the tested compounds was estimated at 50 µg/mL and 100 µg/mL. The red line was equal to 5% of acceptable hemoglobin release. The results were compared with red blood cells treated with PBS buffer (1% of hemolysis—negative control) and 1% SDS (100% of hemolysis—positive control).

**Figure 9 nanomaterials-12-00077-f009:**
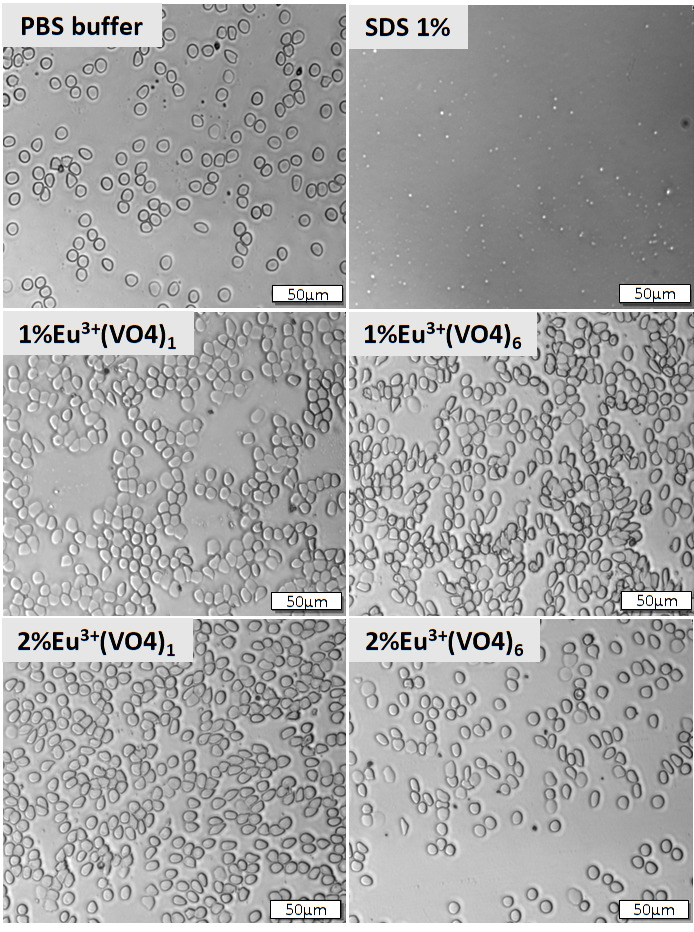
Red blood cell smear was performed with the use of purified sheep erythrocytes. For this experiment, selected compounds were used from the first series: Ca_9.9_Eu_0.1_(PO_4_)_5_(VO_4_)_1_(OH)_2_, Ca_9.9_Eu_0.1_(VO_4_)_6_(OH)_2_ and from the second series: Ca_9.8_Eu_0.2_(PO_4_)_5_(VO_4_)_1_(OH)_2_, Ca_9.8_Eu_0.1_(VO_4_)_6_(OH)_2_. The final concentration of the tested compounds was estimated at 100 µg/mL. The morphology of red blood cells treated with the tested compounds was compared with that of red blood cells treated with PBS buffer (negative control) and 1% SDS (positive control).

**Figure 10 nanomaterials-12-00077-f010:**
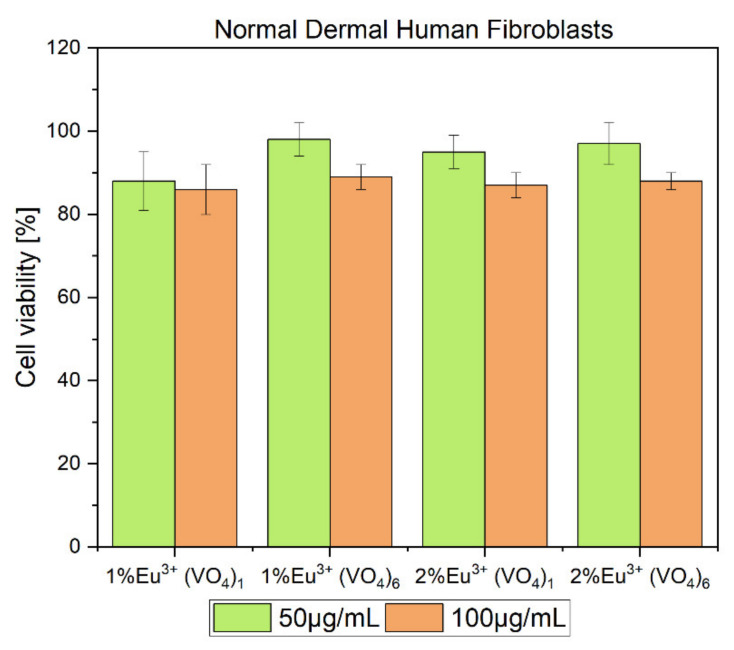
MTT cytotoxicity assay of compounds selected from the first series: Ca_9.9_Eu_0.1_(PO_4_)_5_(VO_4_)_1_(OH)_2_,Ca_9.9_Eu_0.1_(VO_4_)_6_(OH)_2_ and second series: Ca_9.8_Eu_0.2_(PO_4_)_5_(VO_4_)_1_(OH)_2_,Ca_9.8_Eu_0.1_(VO_4_)_6_(OH)_2_. MTT assay was performed by using the NHDF cell line, the final concentration of the tested compounds in culture medium was estimated at 50 µg/mL and 100 µg/mL.

**Table 1 nanomaterials-12-00077-t001:** The element contents in the Ca_9.8_Eu_0.2_(PO_4_)_5_(VO_4_)_x_(OH)_2_ (where x 1–3) based on the ICP-OES technique.

Sample	ICP OES Technique Results			
	n P [mol]	n Ca [mol]	n V [mol]	n Eu [mol]
Ca_9.8_Eu_0.2_(PO_4_)_5_(VO_4_)_1_(OH)_2_	6.35	9.75	1.30	0.25
Ca_9.8_Eu_0.2_(PO_4_)_4_(VO_4_)_2_(OH)_2_	4.32	9.72	0.68	0.28
Ca_9.8_Eu_0.2_(PO_4_)_3_(VO_4_)_3_(OH)_2_	5.37	9.74	0.45	0.26

**Table 2 nanomaterials-12-00077-t002:** The Ca, P, Eu, and V ion release from Ca_9.9_Eu_0.1_(PO_4_)_5_(VO_4_)_1_(OH)_2_ to the simulated body fluid after 0 min, 5 min, 15 min, 30 min, 45 min, 60 min, 360 min, and 1440 min of incubation with simultaneous rotation.

Time of Incubation (min)	Ca	P	Eu	V
0	11 ± 0.5 ppm	2.45 ± 0.1 ppm		
5	11.4 ± 0.5 ppm	2.6 ± 0.1 ppm		
15	12.32 ± 0.5 ppm	2.73 ± 0.1 ppm		
30	12.64 ± 0.6 ppm	2.758 ± 0.1 ppm		
45	12.87 ± 0.6 ppm	2.77 ± 0.1 ppm		0.266 ± 0.01 ppm
60	13.34 ± 0.7 ppm	2.806 ± 0.15 ppm		0.315 ± 0.015 ppm
360	13.58 ± 0.7 ppm	2.8 ± 0.15 ppm		0.543 ± 0.015 ppm
1440	13.84 ± 0.7 ppm	2.8 ± 0.15 ppm		1.227 ± 0.06 ppm

**Table 3 nanomaterials-12-00077-t003:** The Ca, P, Eu, and V ion release from Ca_9.8_Eu_0,2_(PO_4_)_5_(VO_4_)_1_(OH)_2_ to the simulated body fluid after 0 min, 5 min, 15 min, 30 min, 45 min, 60 min, 360 min, and 1440 min of incubation with simultaneous rotation.

Time of Incubation (min)	Ca	P	Eu	V
0	11 ± 0.5 ppm	2.84 ± 0.15 ppm		
5	11.3 ± 0.5 ppm	2.846 ± 0.15 ppm		
15	12.12 ± 0.6 ppm	2.897 ± 0.15 ppm		
30	12.24 ± 0.6 ppm	2.894 ± 0.15 ppm	<0.1 ± 0 ppm	0.2 ± 0.01 ppm
45	12.71 ± 0.6 ppm	2.9 ± 0.15 ppm	<0.1 ± 0 ppm	0.286 ± 0.015 ppm
60	12.77 ± 0.6 ppm	2.9 ± 0.15 ppm	<0.1± 0 ppm	0.351 ± 0.02 ppm
360	13.94 ± 0.7 ppm	2.903 ± 0.15 ppm	<0.1 ± 0 ppm	0.657 ± 0.03 ppm
1440	14.9 ± 0.7 ppm	2.9 ± 0H.15 ppm	<0.1± 0 ppm	1.291 ± 0.06 ppm

## Data Availability

Data are available from the authors upon request.

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
