# Peer review of "A Study of Vanadate Group Substitution into Nanosized Hydroxyapatite Doped with Eu3+ Ions as a Potential Tissue Replacement Material"

_nanomaterials, 2021, doi:10.3390/nano12010077_

Round 1
Reviewer 1 Report
Report on “A study of vanadate group substitution into nanosized hydroxyapatite doped with Eu3+ ion as a potential tissue replacement material”.
This manuscript presents the effects of europium and vanadate ions when inserted in the synthesis procedure of hydroxiapatite. The structural, luminescence and biological properties are investigated.
Here some critical points are reported:
1- line 39- the introduction on bone structure is slightly out of focus and could be cut, otherwise it should be revised: osteons are present only in compact bones, whereas trabecular bone is different.
2- line 99- please check the chemical formula
3- lines 109 and following- The description of experimental procedure is approximate and does not allow reproduction of the experiment. Please add details: reactant amounts, volumes and so concentrations, procedures of addition (and addition rates)...
4- line 112- Is really pH adjusted only after all reactants are put together in the reaction vessel?
5- line 115- Please explain how pH 7 is obtained
6- lines 125-126- results for ICP analyses are not reported anywhere in the text
7- lines 214-215- figure 1 clearly shows the presence of a secondary phase already when vanadate is 2. The presence of peaks not belonging to hydroxyapatite increase on increasing vanadate concentration. Pyrovanadate peaks in the range 25-30° prevail at high V content but are appreciable immediately after any V insertion. Please deeply revise all the XRD analysis results.
8- line 222- the substitution of vanadate in the place of phosphate must be supported by results: please provide structural-crystallographic study with evidence for substitution.
9- lines 236-238- If the same pH value is used, please explain the mechanism by which the use of NH3 or NaOH should change synthesis results.
10- figure 2- one of the 2 images is redundant (they show almost the same range with the same magnification): please use only one for the sake of clarity
11- line 265- please check the formula
12- table 1- EDS analysis is an elemental analysis. Please delete the charges, because ions indication is not appropriate
13- figure 3- never cited in the text. Furthermore SEM images show magnifications that do not allow to see crystals and are not meaningful. Please show images at higher magnification
14- line 292- EDS analyses is not quantitative, but only indicates the presence of elements with large approximation. Please report ICP results: this technique is much more accurate and reliable
15- line 293- considering the behavior of elements and looking at all related literature, it is hardly believable there there is a quantitative incorporation of Eu and vanadate into hydroxyapatite structure. Also see comment #14
16- line 306-
- please explain the rational of studying the photophysical properties of materials that are mixtures of crystalline phases (high amount of V, which implies the presence of pyrovanadate)
- please quantify the presence of different crystallographic phases in all the samples
17- Line 328 - please indicate the 3 different crystallographic sites
18- line 364- this number has not been calculated, please add this calculation, otherwise all the study results only qualitative
19- line 375- this discussion is very qualitative, not supported by data
20- line 456- conclusion not supported by data
21- line 457-459- this sentence is obscure
22- line 461- doped or substituted? the 2 terms are deeply different
Author Response
Dear Editor,
We would like to express our sincerest gratitude to the Reviewer for the enormous efforts in criticizing the manuscript. We have taken into account all raised question here follows the detailed answers to the Reviewer. All changes we have made to the original manuscript, are marked in the red color in the text.
Reviewer 1
Q 1: line 39- the introduction on bone structure is slightly out of focus and could be cut, otherwise it should be revised: osteons are present only in compact bones, whereas trabecular bone is different.
Response 1: We would like to thank for the suggestion. We have decided to remove this sentence from this part and improve introduction. - Line 39
Q 2: line 99- please check the chemical formula
Response 2: We would like to thank for the suggestion. We have corrected the chemical formula. – Line 112
Q 3: lines 109 and following- The description of experimental procedure is approximate and does not allow reproduction of the experiment. Please add details: reactant amounts, volumes and so concentrations, procedures of addition (and addition rates)...
Response 3: We have improved our manuscript and added the necessary information to reproduce the experiments. – line 122
Q 4: line 112- Is really pH adjusted only after all reactants are put together in the reaction vessel?
Response 4: Yes, pH is adjusted after all substrates that are mixed together. – Line 130
Q 5: line 115- Please explain how pH 7 is obtained
Response 5: We have added the explanation. – Line 132
Q 6: lines 125-126- results for ICP analyses are not reported anywhere in the text
Response 6: We have improved our manuscript and added the ICP OES results – Line 336
Q 7: lines 214-215- figure 1 clearly shows the presence of a secondary phase already when vanadate is 2. The presence of peaks not belonging to hydroxyapatite increase on increasing vanadate concentration. Pyrovanadate peaks in the range 25-30° prevail at high V content but are appreciable immediately after any V insertion. Please deeply revise all the XRD analysis results.
Response 7: We have corrected our discussion, - Line 267
Q 8: line 222- the substitution of vanadate in the place of phosphate must be supported by results: please provide structural-crystallographic study with evidence for substitution.
Response 8: We have added the extra results of structural-crystallographic studies – please see Supplementary data.
Q 9: lines 236-238- If the same pH value is used, please explain the mechanism by which the use of NH3 or NaOH should change synthesis results.
Response 9: It has been suggested that some data might indicate the use of NaOH is more suitable for the substituted hydroxyapatite with vanadate groups. In this case, the Discussion has been supported with new references. – Line 296
Q 10: figure 2- one of the 2 images is redundant (they show almost the same range with the same magnification): please use only one for the sake of clarity
Response 10: We have corrected the Figure 2 and provided only one images. – Line 317
Q 11: line 265- please check the formula
Response 11: We have checked the formula and corrected it. – Line 333
Q 12: table 1- EDS analysis is an elemental analysis. Please delete the charges, because ions indication is not appropriate
Response 12: We have corrected this mistake and removed the ion indication. – Line 346
Q 13: figure 3- never cited in the text. Furthermore SEM images show magnifications that do not allow to see crystals and are not meaningful. Please show images at higher magnification
Response 13: We would like to kindly thank you for this suggestion. We did provide description of Figure 3 in the text. – Line 356. Also we did improve the SEM-EDS images. – Line 351
Q 14: line 292- EDS analyses is not quantitative, but only indicates the presence of elements with large approximation. Please report ICP results: this technique is much more accurate and reliable
Response 14: It has been corrected and added more accurate data related to ICP OES measurement. – Line 363 and Line 364
Q 15: line 293- considering the behavior of elements and looking at all related literature, it is hardly believable there is a quantitative incorporation of Eu and vanadate into hydroxyapatite structure. Also see comment #14
Response 15: It has been added ICP OES measurement. It is well known that is difficult to incorporate vanadium ions into hydroxyapatite matrix. However, Eu3+ ions can be easily incorporated into hydroxyapatite crystal matrix as the result of their similar ionic radius to calcium ions (Kalaivani S , Kannan S . Collective substitutions of selective rare earths (Yb3+, Dy3+, Tb3+, Gd3+, Eu3+, Nd3+) in ZrO2: an exciting prospect for biomedical applications. Dalton Trans. 2019 Jun 25;48(25):9291-9302. doi: 10.1039/c9dt01930h. PMID: 31166338.) – Line 364
Q 16: line 306-
- please explain the rational of studying the photophysical properties of materials that are mixtures of crystalline phases (high amount of V, which implies the presence of pyrovanadate)
- please quantify the presence of different crystallographic phases in all the samples
Response 16:
- It has been explained the experiment in the text. – Line 373
- It has been provided the Rietveld calculation – Supplementary data.
Q 17: Line 328 - please indicate the 3 different crystallographic sites
Response 17: It has been provided such information. – Line 417
Q 18: line 364- this number has not been calculated, please add this calculation, otherwise all the study results only qualitative
Response 18: It has been provided the qualitative data.
Q 19: line 375- this discussion is very qualitative, not supported by data
Response 19: It has been added the proper references in the Discussion. – Line 462 and Line 467
Q 20: line 456- conclusion not supported by data
Response 20: Now, the Conclusion contains only the summary of the obtained results. Moreover, the obtained results have been described and discussed in the Results and discussion section, as well.
Q 21: line 457-459- this sentence is obscure
Response 21: It has been corrected. – Line 578
Q 22: line 461- doped or substituted? the 2 terms are deeply different
Response 22: This mistake has been corrected. – Line 582

Reviewer 2 Report
General comments
The present manuscript reports on the production and characterisation of nanosized vanadate-substituted hydroxyapatites doped with Eu3+ ions via the precipitation method.
The topic well fits the aim and scope of Nanomaterials, but it has to be strongly improved before considering it for publication.
Specific remarks are reported below, point by point.
Abstract
- A contextualization has to be added as incipit of the Abstract section.
Keywords
One keyword about the potential application has to be added at the end of the keywords list.
- Introduction
- The introduction has to be improved, adding more references and evidencing the purpose and the originality of the present work.
- The following period “Bone injuries, especially in the case of severe fractures, are a problem not only in the regeneration of the bone tissue itself but also in the tissues adjacent to the damaged bone, such as cartilage, muscle or nervous tissue. Regeneration and healing of bone tissue is the physiological process of body repair. In optimal conditions repair processes can lead to complete renewal of bone structure, however, the repair capacity of the nervous tissue is quite limited. It is associated with a long recovery process and often does not bring the desired effects, especially if the damaged structures, such as the skull and spine, protect the most important parts of the human body—the brain and spinal cord” needs suitable references.
- The statements “After its success in the field of physical, chemical, and medical sciences, nanotechnology has now started revolutionizing the bio-detection and drug delivery sciences and bio regeneration techniques. The specific advantages include superior pharmacodynamics, pharmacokinetics, reduced toxicity, and targeting capability” have to be supported with proper references.
- Similarly, the sentences “Hydroxyapatites are well-known for their biocompatibility and osteoinductivity properties towards living cells, especially bone tissue. Synthetic apatite has a highly developed specific surface and therefore it can form a strong bond with living tissue, such as bone and dental tissue” have to be corroborated with appropriate references, including “Multisubstituted hydroxyapatite powders and coatings: the influence of the codoping on the hydroxyapatite performances, Journal of Applied Ceramic Technology 6 (2019):1864–1884“ and ”Cationic and Anionic substitutions in hydroxyapatite, In: Handbook of Bioceramics and Biocomposites, Springer International Publishing 2016: 145-211.”.
- More publications have to be cited for the following considerations “It has been confirmed by many research studies that especially hydroxyapatite-based scaffolds when combined with biodegradable polymer biomaterials such as collagen, polylactide acid, chitosan or alginate, produced promising effects due to their ability to act with living tissue, high biocompatibility and induction of cell proliferation and growth of bone tissue [3,5,10,11].”. Moreover, some of the cited ones are not appropriate and can be replaced with more proper ones, including “Biosynthesis of innovative calcium phosphate/hydrogel composites: physicochemical and biological characterisation. Nanotechnology 32(9) (2021): 095102” and “Poly(L-lactic acid)/calcium-deficient nanohydroxyapatite electrospun mats for murine bone marrow stem cell cultures, Journal of Bioactive and Compatible Polymers 26[3] (2011): 225-241”.
- The Authors have to support the following phrase “Vanadium and vanadium compounds are commonly used in industry as a drying agent in paints, as a photographic developer, it is also used in black dyes, inks or pigments for textiles, ceramics and printings” with suitable references.
- The potential applications have to be more evidenced.
- The Introduction section seems not completed. At the end, the aim of the paper has to be highlighted, as well as the originality and added value with respect to previous papers about the same topic.
- It is strongly suggested to add a brief list of the used characterisations at the end of the Introduction section.
- Materials and Methods
2.1. Synthesis method
- The concentrations of all reactants have to be specified.
- Were the solutions mixed by dropping? Please specify the dropping rate.
- The stirring rate has to be specified.
- The hating and cooling rates for the calcination have to be indicated.
2.2. Physicochemical characterization
2.2.1. Material characterization
- For XRD analysis, the time per step and step size have to be reported.
- More details about the ICP analysis have to be added.
- For ATR, the resolution and the scans number have to be specified.
- For EDS, it is not correct to affirm that it is necessary to establish the chemical formula. It is not a quantitative analysis.
- For SEM, details about the samples preparation for observation have to be introduced.
2.2.2. Luminescence properties
- The statement “The excitation spectra and luminescence kinetics were recorded at 618 nm according to the most intense electric dipole transition (from level 5D0 → 7F2 level) and excited at 396 nm” has to be supported with proper literature references, including “Synthesis, thermal behaviour and luminescent properties of rare earth-doped titania nanofibers, Chemical Engineering Journal 166[2] (2011): 751-764”.
2.3. Evaluation of biocompatibility
2.3.1. Preparation of nanosized vanadium-substituted hydroxyapatite suspension
- The suspensions concentrations have to be specified.
- Results and discussion
3.1. Analysis of structure and morphology
- The Authors stated that “For the above-mentioned powders, signals in the range of 31° to 34° in the experimental patterns correspond to distinctive phosphate groups of hydroxyapatite crystal structure”, but it is not correct. They represent typical crystal planes in the apatite lattice.
- Moreover, the Authors talked of broader bases of the peaks that indicate the nanosized structure of the nanosized materials, but they could evaluate the crystallite dimensions using the Scherrer formula, in order to provide some numbers and to compare them.
- They could also estimate the crystallinity degree using the relation reported in “F-substituted hydroxyapatite nanopowders: thermal stability, sintering behavior and mechanical properties, Ceramics International 36[1] (2010): 313-322”, properly citing this reference.
- The following statement “The gradual increase in the vanadate groups eventually leads to the gradual decrease of the intensity of signal from phosphate groups and increase of the intensity of signal from vanadate groups” is no sense…it is not clear. Please clarify this point.
- The following considerations “Nonetheless, there are also some data which indicate that the ammonia environment is not suitable to obtain vanadate-substituted hydroxyapatite. To obtain an alkaline environment of a chemical reaction during the synthesis of vanadate-substituted hydroxyapatite, NaOH should be substituted for NH3∙H2O. Moreover, other substrates can be used during the synthesis of vanadate-substituted hydroxyapatite.” need proper references.
- In Fig 1 the XRD peaks have to be clearly assigned.
- All the reported FTIR peaks assignments, such as “The absorption bands of the phosphate group at 560.70 cm-1 and 600.24 cm-1 correspond to the double degenerate bending mode (ν2) of the P-O-P bonds and triply degenerate bending 255 mode (ν4) of the P-O bonds, respectively” have to be supported with related references, including “Thermal stability and sintering behaviour of hydroxyapatite nanopowders, J. Thermal Anal. Calor. 88 (2007): 237-243.”.
- The experimental evidence “It is also noticeable (Figure 2) that the additional incorporation of (VO43-) groups into the hydroxyapatite framework, unalterably entails the shift of the absorption bands towards lower wavelength” has to be corroborated with appropriate references.
- The following statement “Gradual increment of the number of vanadium groups substituted for phosphate groups leads to the appearance of characteristic vibrational bands which invoke the appearance of Ca2O2V7 crystal structure” has to be justified with proper references.
- - The following assignments “Typical vibrational bands at 417.03 cm-1 (ν4) and 561.66 cm-1 (ν3) refer to the asymmetric vibration models of O-Ca-O and O-V-O bending, respectively. The vibrational bands which are noticeable at 825.38 cm-1 (ν1) and 760.78 cm-1 (ν2) correspond to the stretching frequencies of the V-O group” have to be corroborated with proper references.
- Similarly the sentences “The described characteristic bonding of O-Ca-O, O-V-O and V-O groups is the most visible for the sample material with the greatest numbers of vanadium groups (VO43-). The fewer vanadium groups are incorporated into hydroxyapatite structure, the smaller band vibrations occur; additionally, the characteristic shift into the direction of hydroxyapatite is visible and typical phosphate groups (PO43-) can be noticed” have to be supported with literature references.
3.2. Luminescence properties
- The following assignments “Five typical transitions of Eu3+ ions are present in the spectra, at wavelengths of 575 nm, 585 nm, 618 nm, 660 nm and 710 nm, which correspond to the transition from the excited level of 5D0 to the levels of 7F0-4. The transitions 312 are assigned as 5D0→7F0, 5D0→7F1, 5D0→7F2, 5D0→7F3 and 5D0→7F4, respectively, with increasing wavelength value. The most intense peak corresponds to the 5D0→7F2 transition, for which emission is observed at wavelengths in the range of 600-625 nm, while the maximum intensity is observed at 618 nm” have to be supported with proper references, including “Eu-doped titania nanofibers: processing, thermal behaviour and luminescent properties, Journal of Nanoscience and Nanotechnology 10 (2010): 5183-5190”.
- Similarly, the assignment “In (VO43-)x spectra, where x ranges from 1 to 5, apart from the transition’s characteristic for Eu3+ ions, an intense peak is visible at a wavelength of approximately 270 nm, corresponding to the charge transfer of an electron between the ionized oxygen atom and the europium ion (O2-→Eu3+).” needs suitable references.
- The Authors have to corroborate the consideratiosn “Moreover, the charge transfer from oxygen to vanadium seems to be slightly shifted toward the highest wavelength number, however, it can be caused by the incorporation of the vanadium groups into the hydroxyapatite framework. Nevertheless, the peak positions for the excitation and emission spectra are in agreement with those expected for Eu3+ ions incorporated in exchange for calcium (II) ions into the hydroxyapatite structure.” with appropriate references.
3.3. Biological properties
- The consideration “However, on the other hand, there is much evidence that indicates the positive influence on living organisms, for example neuroprotective and neuroregenerative properties” needs to be supported with appropriate references.
Author Response
Dear Editor,
We would like to express our sincerest gratitude to the Reviewer for the enormous efforts in criticizing the manuscript. We have taken into account all raised question here follows the detailed answers to the Reviewer. All changes we have made to the original manuscript, are marked in the red color in the text.
Reviewer 2
Q 1: Abstract A contextualization has to be added as incipit of the Abstract section.
Response 1: It has been added the contextualization in the end of the Introduction section – Line 103
Q 2: Keywords One keyword about the potential application has to be added at the end of the keywords list
Response 2: It has been added the keyword about the potential application. – Line 24
Q 3: Introduction The introduction has to be improved, adding more references and evidencing the purpose and the originality of the present work.
Response 3: The Introduction has been improved and there have been added the references to support our research.
Q 4: Introduction The following period “Bone injuries, especially in the case of severe fractures, are a problem not only in the regeneration of the bone tissue itself but also in the tissues adjacent to the damaged bone, such as cartilage, muscle or nervous tissue. Regeneration and healing of bone tissue is the physiological process of body repair. In optimal conditions repair processes can lead to complete renewal of bone structure, however, the repair capacity of the nervous tissue is quite limited. It is associated with a long recovery process and often does not bring the desired effects, especially if the damaged structures, such as the skull and spine, protect the most important parts of the human body—the brain and spinal cord” needs suitable references.
Response 4: The manuscript has been improved. The new references have been added. – Line 38
Q 5: Introduction The statements “After its success in the field of physical, chemical, and medical sciences, nanotechnology has now started revolutionizing the bio-detection and drug delivery sciences and bio regeneration techniques. The specific advantages include superior pharmacodynamics, pharmacokinetics, reduced toxicity, and targeting capability” have to be supported with proper references.
Response 5: The manuscript has been improved. The new references have been added. – Line 49
Q 6: Introduction Similarly, the sentences “Hydroxyapatites are well-known for their biocompatibility and osteoinductivity properties towards living cells, especially bone tissue. Synthetic apatite has a highly developed specific surface and therefore it can form a strong bond with living tissue, such as bone and dental tissue” have to be corroborated with appropriate references, including “Multisubstituted hydroxyapatite powders and coatings: the influence of the codoping on the hydroxyapatite performances, Journal of Applied Ceramic Technology 6 (2019):1864–1884“ and ”Cationic and Anionic substitutions in hydroxyapatite, In: Handbook of Bioceramics and Biocomposites, Springer International Publishing 2016: 145-211.”
Response 6: The sentence has been corrected. – Line 52
Q 7: Introduction More publications have to be cited for the following considerations “It has been confirmed by many research studies that especially hydroxyapatite-based scaffolds when combined with biodegradable polymer biomaterials such as collagen, polylactide acid, chitosan or alginate, produced promising effects due to their ability to act with living tissue, high biocompatibility and induction of cell proliferation and growth of bone tissue [3,5,10,11].”. Moreover, some of the cited ones are not appropriate and can be replaced with more proper ones, including “Biosynthesis of innovative calcium phosphate/hydrogel composites: physicochemical and biological characterisation. Nanotechnology 32(9) (2021): 095102” and “Poly(L-lactic acid)/calcium-deficient nanohydroxyapatite electrospun mats for murine bone marrow stem cell cultures, Journal of Bioactive and Compatible Polymers 26[3] (2011): 225-241”.
Response 7: Thank you very much for Qing this out. We did improve our manuscript and add the necessary references. – Line 59
Q 8: Introduction The Authors have to support the following phrase “Vanadium and vanadium compounds are commonly used in industry as a drying agent in paints, as a photographic developer, it is also used in black dyes, inks or pigments for textiles, ceramics and printings” with suitable references.
Response 8: The manuscript has been improved. The new references have been added. – Line 83
Q 9: Introduction The potential applications have to be more evidenced.
Response 9: The new references have been added. – Line 98
Q 10: Introduction The Introduction section seems not completed. At the end, the aim of the paper has to be highlighted, as well as the originality and added value with respect to previous papers about the same topic.
Response 10: The aim of the study has been added.– Line 103
Q 11: Introduction It is strongly suggested to add a brief list of the used characterisations at the end of the Introduction section.
Response 11: The list of used characterisation has been added in the end of the Abstract. – Line 23
Q 12: Synthesis method. The concentrations of all reactants have to be specified.
Response 12: Such information has been added. – Line 115
Q 13: Synthesis method. Were the solutions mixed by dropping? Please specify the dropping rate.
Response 13: The reagents have been added one to another by gently pouring, not adding drop by drop.
Q 14: Synthesis method. The stirring rate has to be specified.
Response 14: This information has been provided. – Line 130
Q 15: Synthesis method. The heating and cooling rates for the calcination have to be indicated.
Response 15: This information has been provided. – Line 133
Q 16: Material characterization. For XRD analysis, the time per step and step size have to be reported.
Response 16: This information has been added. – Line 141
Q 17: Material characterization. More details about the ICP analysis have to be added.
Response 17: This information has been added. – Line 144
Q 18: Material characterization. For ATR, the resolution and the scans number have to be specified.
Response 18: This information has been added.. – Line 153
Q 19: Material characterization. For EDS, it is not correct to affirm that it is necessary to establish the chemical formula. It is not a quantitative analysis.
Response 19: This information has been corrected. – Line 157
Q 20: Material characterization. For SEM, details about the samples preparation for observation have to be introduced.
Response 20: This information has been added. – Line 163
Q 21: Luminescence properties. The statement “The excitation spectra and luminescence kinetics were recorded at 618 nm according to the most intense electric dipole transition (from level 5D0 → 7F2 level) and excited at 396 nm” has to be supported with proper literature references, including “Synthesis, thermal behaviour and luminescent properties of rare earth-doped titania nanofibers, Chemical Engineering Journal 166[2] (2011): 751-764”.
Response 21: The manuscript has been improved. The new references have been added. – Line 178
Q 22: Preparation of nanosized vanadium-substituted hydroxyapatite suspension The suspensions concentrations have to be specified.
Response 22: This information has been added. – Line 182
Q 23: Analysis of structure and morphology. The Authors stated that “For the above-mentioned powders, signals in the range of 31° to 34° in the experimental patterns correspond to distinctive phosphate groups of hydroxyapatite crystal structure”, but it is not correct. They represent typical crystal planes in the apatite lattice.
Response 23: According to the ICSD 262004 reference for hydroxyapatite, the range from 31° to 34° is also characteristic for the hydroxyapatite structure.
Q 24: Analysis of structure and morphology. Moreover, the Authors talked of broader bases of the peaks that indicate the nanosized structure of the nanosized materials, but they could evaluate the crystallite dimensions using the Scherrer formula, in order to provide some numbers and to compare them.
Response 24: The new references have been added. – Line 277
Q 25: Analysis of structure and morphology. They could also estimate the crystallinity degree using the relation reported in “F-substituted hydroxyapatite nanopowders: thermal stability, sintering behavior and mechanical properties, Ceramics International 36[1] (2010): 313-322”, properly citing this reference.
Response 25: Thank you very this suggestion but evaluating of the crystallinity degree is not the aim of our research.
Q 26: Analysis of structure and morphology. The following statement “The gradual increase in the vanadate groups eventually leads to the gradual decrease of the intensity of signal from phosphate groups and increase of the intensity of signal from vanadate groups” is no sense…it is not clear. Please clarify this Q.
Response 26: The sentence has been clarified. – Line 277
Q 27: Analysis of structure and morphology. The following considerations “Nonetheless, there are also some data which indicate that the ammonia environment is not suitable to obtain vanadate-substituted hydroxyapatite. To obtain an alkaline environment of a chemical reaction during the synthesis of vanadate-substituted hydroxyapatite, NaOH should be substituted for NH3∙H2O. Moreover, other substrates can be used during the synthesis of vanadate-substituted hydroxyapatite.” need proper references.
Response 27: The manuscript has been improved. The new references have been added. – Line 295
Q 28: Analysis of structure and morphology. In Fig 1 the XRD peaks have to be clearly assigned.
Response 28: The Figure 1. has been improved by adding labels for hydroxyapatite and calcium pyrovanadate. – Line 314
Q 29: Analysis of structure and morphology. All the reported FTIR peaks assignments, such as “The absorption bands of the phosphate group at 560.70 cm-1 and 600.24 cm-1 correspond to the double degenerate bending mode (ν2) of the P-O-P bonds and triply degenerate bending 255 mode (ν4) of the P-O bonds, respectively” have to be supported with related references, including “Thermal stability and sintering behaviour of hydroxyapatite nanopowders, J. Thermal Anal. Calor. 88 (2007): 237-243.”.
- The experimental evidence “It is also noticeable (Figure 2) that the additional incorporation of (VO43-) groups into the hydroxyapatite framework, unalterably entails the shift of the absorption bands towards lower wavelength” has to be corroborated with appropriate references.
- The following statement “Gradual increment of the number of vanadium groups substituted for phosphate groups leads to the appearance of characteristic vibrational bands which invoke the appearance of Ca2O2V7 crystal structure” has to be justified with proper references.
- - The following assignments “Typical vibrational bands at 417.03 cm-1 (ν4) and 561.66 cm-1 (ν3) refer to the asymmetric vibration models of O-Ca-O and O-V-O bending, respectively. The vibrational bands which are noticeable at 825.38 cm-1 (ν1) and 760.78 cm-1 (ν2) correspond to the stretching frequencies of the V-O group” have to be corroborated with proper references.
- Similarly the sentences “The described characteristic bonding of O-Ca-O, O-V-O and V-O groups is the most visible for the sample material with the greatest numbers of vanadium groups (VO43-). The fewer vanadium groups are incorporated into hydroxyapatite structure, the smaller band vibrations occur; additionally, the characteristic shift into the direction of hydroxyapatite is visible and typical phosphate groups (PO43-) can be noticed” have to be supported with literature references.
Response 29: The manuscript has been improved. The new references have been added:
– Line 331
– Line 333
– Line 338
– Line 344
Q 30: Luminescence properties. The following assignments “Five typical transitions of Eu3+ ions are present in the spectra, at wavelengths of 575 nm, 585 nm, 618 nm, 660 nm and 710 nm, which correspond to the transition from the excited level of 5D0 to the levels of 7F0-4. The transitions 312 are assigned as 5D0→7F0, 5D0→7F1, 5D0→7F2, 5D0→7F3 and 5D0→7F4, respectively, with increasing wavelength value. The most intense peak corresponds to the 5D0→7F2 transition, for which emission is observed at wavelengths in the range of 600-625 nm, while the maximum intensity is observed at 618 nm” have to be supported with proper references, including “Eu-doped titania nanofibers: processing, thermal behaviour and luminescent properties, Journal of Nanoscience and Nanotechnology 10 (2010): 5183-5190”.
Response 30: The manuscript has been improved. The new references have been added. – Line 404
Q 31: Luminescence properties Similarly, the assignment “In (VO43-)x spectra, where x ranges from 1 to 5, apart from the transition’s characteristic for Eu3+ ions, an intense peak is visible at a wavelength of approximately 270 nm, corresponding to the charge transfer of an electron between the ionized oxygen atom and the europium ion (O2-→Eu3+).” needs suitable references.
Response 31: The manuscript has been improved. The new references have been added. – Line 462
Q 32: Luminescence properties The Authors have to corroborate the consideratiosn “Moreover, the charge transfer from oxygen to vanadium seems to be slightly shifted toward the highest wavelength number, however, it can be caused by the incorporation of the vanadium groups into the hydroxyapatite framework. Nevertheless, the peak positions for the excitation and emission spectra are in agreement with those expected for Eu3+ ions incorporated in exchange for calcium (II) ions into the hydroxyapatite structure.” with appropriate references.
Response 32: The manuscript has been improved. The new references have been added. – Line 467
Q 33: Biological properties. The consideration “However, on the other hand, there is much evidence that indicates the positive influence on living organisms, for example neuroprotective and neuroregenerative properties” needs to be supported with appropriate references.
Response 33: The manuscript has been improved. The new references have been added. – Line 531

Reviewer 3 Report
In this study, the authors prepared and investigated the Eu-containing vanadate-substituted hydroxyapatites. In general, the article can be reconsidered for publication after a major revision.
1) The purpose of this work is not clear and must be indicated at the end of the Introduction part.
2) XRD analysis - label the peaks that belong to HAP and Ca2V2O7. What was the % of Ca2V2O7 in HAP?
3) QY of prepared samples should be measured and compared. Later, the authors can use the optimal sample for biological activity testing.
4) It is known that prolonged heavy metal release is harmful to live organisms. Hence, Eu and V release in SBF should be tested over several days.
5) Fluorescence images of cells incubated with the sample can be shown - to confirm that the optimal sample can label cells (red color).
6) The Introduction part can be improved - Eu-containing compounds are not only fluorescent but also served as good X-ray attenuation agents for CT scanning. The following works can be referred for example DOI: 10.1088/2632-959X/abe343 and DOI: 10.2174/1389557520666200604163452
Author Response
Dear Editor,
We would like to express our sincerest gratitude to the Reviewer for the enormous efforts in criticizing the manuscript. We have taken into account all raised question here follows the detailed answers to the Reviewer. All changes we have made to the original manuscript, are marked in the red color in the text.
Reviewer 3
Q 1: The purpose of this work is not clear and must be indicated at the end of the Introduction part.
Response 1: The manuscript has been improved in this particular section.– Line 103
Q 2: XRD analysis - label the peaks that belong to HAP and Ca2V2O7. What was the % of Ca2V2O7 in HAP
Response 2: Thank you for the suggestion. Indeed we did label ranges for hydroxyapatite and calcium pyrovanadate. – Line 314
Q 3: QY prepared samples should be measured and compared. Later, the authors can use the optimal sample for biological activity testing.
Response 3: Thank you very much for this suggestion. The absolute quantum efficiency was measured with the help of the Hamamatsu Absolute PL quantum yields measurement system C9920-02G. Unfortunately, the results of the QY for the obtained samples were insufficient. It has been decided to not add this data.
Q 4: It is known that prolonged heavy metal release is harmful to live organisms. Hence, Eu and V release in SBF should be tested over several days
Response 4: Such experiment has been provided. – Line 238 and 540
Q 5: Fluorescence images of cells incubated with the sample can be shown - to confirm that the optimal sample can label cells (red color).
Response 5: Thank you for this suggestion. However, it can be difficult but not impossible to use the obtained materials as bio-labels.
Q 6: The Introduction part can be improved - Eu-containing compounds are not only fluorescent but also served as good X-ray attenuation agents for CT scanning. The following works can be referred for example DOI: 10.1088/2632-959X/abe343 and DOI: 10.2174/1389557520666200604163452
Response 6: The suggested works have been added. – Line 82

Round 2
Reviewer 1 Report
1- Line 111- “where x = 1.0, 2.0, mol% and y = 1, 2, 3, 4, 5, 6” has no chemical meaning. Please check and correct
2- Table 1 – Authors consider obtained phases as stoichiometric substituted hydroxyapatites. Furthermore the formulas indicate that both vanadate and Eu ion substitute for anions and cations into the structure of the stoichiometric htdroxyapatite, respectively. This can not be correct, since the patterns shown in figure 1 display broad peaks and indicate that the obtained materials are poorly crystalline.
3- Why Eu is considered as substituion in the structure? How is it possible to calculate structural parameters that identify the contribution of Eu in such a low percentage? And furthermore in the presence of a supposed multiple ionic substitution?
4- Table S1- Please comment these data indicating the different contribution of 1-Eu substitution;2-vanadate substitution;3-presence of pyrovanadate.
5- Figure 3- the morphology of cystals is barely appreciable. Higher magnification and resolution is required for SEM. Otherwise TEM analysis is suggested.
Author Response
Dear Editor,
We would like to express our sincerest gratitude to the Reviewer for the enormous efforts in criticizing the manuscript. We have considered all raised question here follows the detailed answers. Moreover, all changes we have made to the original manuscript, are marked in the red color in the text.
Comments and Suggestions for Authors:
Reviewer 1
Q1. Line 111- “where x = 1.0, 2.0, mol% and y = 1, 2, 3, 4, 5, 6” has no chemical meaning. Please check and correct
Response: Thank you very much for pointing out this mistake, we corrected the formula and the sentence. – Line 111
Q2. - Table 1 – Authors consider obtained phases as stoichiometric substituted hydroxyapatites. Furthermore the formulas indicate that both vanadate and Eu ion substitute for anions and cations into the structure of the stoichiometric hydroxyapatite, respectively. This cannot be correct, since the patterns shown in figure 1 display broad peaks and indicate that the obtained materials are poorly crystalline.
Response: It can be confirmed that the obtained hydroxyapatites with one vanadate group were stochiometric and phase pure. The rest were non-stochiometric and phase impure. It was confirmed by Rietveld calculation and the obtained materials are enough crystalline. The peaks are very well formed but broad because the crystallites are small about 25-90 nm. If there were amorphous materials or poor-crystalline materials, there would be observed the broad band in the range form 5 - 30 2θ.
Q3. Why Eu is considered as substitution in the structure? How is it possible to calculate structural parameters that identify the contribution of Eu in such a low percentage? And furthermore in the presence of a supposed multiple ionic substitution?
Response: The Rietveld calculation/refinement uses a least squares approach to refine a theoretical line profile until it matches the measured profile. Moreover, it can be calculated the structural parameters. The structural refinement method has several advantages over conventional quantitative analysis methods. Because the method uses a whole pattern-fitting algorithm, all lines for each phase are explicitly considered, and even severely overlapped lines are usually not a problem. The quality of structural refinement is generally checked by R-values and these numbers are easy to detect as they are consistent with the structure. However, a difference in plotting observed and calculated patterns is the best way to judge the success of Rietveld refinement. Furthermore, other parameters with additional functions were applied to find a structural refinement with better quality and reliability. The optimized parameters were: scale factor, background with exponential shift, exponential thermal shift and polynomial coefficients, basic phase, microstructure, crystal structure, size strain (anisotropic, no rules), structure solution model (genetic algorithm SDPD), shift lattice constants, profile half-width parameters (u, v, w), texture, lattice parameters (a, b, c), factor occupancies and atomic site occupancies (Wyckoff). The introduction of this technique has been a significant step forward in the diffraction analysis of powder samples because it can be used with strongly overlapping reflections.
In our case, there were measured luminescence properties of the obtained materials and after analysis of the Stark components in the luminescence spectra in correlation with the crystallographic data, it can be identified the contribution of Eu3+ ions into the obtained materials.
Q4. Table S1- Please comment these data indicating the different contribution of 1-Eu substitution;2-vanadate substitution;3-presence of pyrovanadate.
Response The Rietveld refinement uses a least squares approach to refine a theoretical line profile until it matches the measured profile. There can be calculated the structural parameters without taking into account the different contribution of the doping ions or groups. Moreover, the quality of structural refinement is generally checked by R-values.
Q5. Figure 3- the morphology of crystals is barely appreciable. Higher magnification and resolution is required for SEM. Otherwise TEM analysis is suggested.
Response: Thank you for your suggestion. There have been added TEM images in the manuscript.

Reviewer 2 Report
General comments
The Authors did not apply all the requested corrections, as evidenced below.
Abstract
- A contextualization has not been added as incipit of the Abstract section, as requested.
- Introduction
- The aim of the paper has been highlighted, whereas the originality and added value with respect to previous papers about the same topic has not been evidenced.
- A brief list of the used characterisations has not been added at the end of the Introduction section.
- Materials and Methods
2.1. Synthesis method
2.2. Physicochemical characterization
2.2.1. Material characterization
- As already evidenced, for EDS, it is not correct to affirm that it is necessary to establish the chemical formula. It is not a quantitative analysis.
- For samples sputtering, the applied current and deposition time have to be specified.
- Results and discussion
3.1. Analysis of structure and morphology
- As already reported, it is not correct to affirm that “signals in the range of 31° to 34° in the experimental patterns correspond to distinctive phosphate groups of hydroxyapatite crystal structure”, they are typical crystal planes in the calcium phosphate lattice.
- As already requested, the Authors have to report quantitative estimation about the crystallite dimensions using the Scherrer formula.
- If not an aim of the present paper, the hydroxyapatite crystallinity degree is an important information and to estimate it requests 1 minute….You can use the relation reported in “F-substituted hydroxyapatite nanopowders: thermal stability, sintering behavior and mechanical properties, Ceramics International 36[1] (2010): 313-322”, properly citing this reference.
Author Response
Dear Editor,
We would like to express our sincerest gratitude to the Reviewer for the enormous efforts in criticizing the manuscript. We have considered all raised question here follows the detailed answers. Moreover, all changes we have made to the original manuscript, are marked in the red color in the text.
Comments and Suggestions for Authors:
Reviewer 2
Q1. Abstract: A contextualization has not been added as incipit of the Abstract section, as requested.
Response: Generally, the contextualization has been taken into account.
Q2. Introduction: The aim of the paper has been highlighted, whereas the originality and added value with respect to previous papers about the same topic has not been evidenced.
Response: Thank you for this suggestion, it has been explained the originality of our research by the end of the introduction. – Line 103-111
Q3. Introduction: A brief list of the used characterisations has not been added at the end of the Introduction section.
Response: Thank you for pointing that out. According with the MS Word template (https://www.mdpi.com/files/word-templates/nanomaterials-template.dot), it is not allowed to include list of used characterisations at the end of the “Introduction” part. However, it has been presented the list of the keywords by the end of “Abstract” part – Line 23-24
Q4. Material characterization: As already evidenced, for EDS, it is not correct to affirm that it is necessary to establish the chemical formula. It is not a quantitative analysis.
Response: EDX technique is used for the elemental analysis or chemical characterization of samples. However, it has been decided to extend our study about ICP OES spectroscopy that is more precise analytical technique used for the detection of chemical elements and the elemental analysis. Line: 351-352
Q5. Material characterization: For samples sputtering, the applied current and deposition time have to be specified.
Response: The concentrations of Eu, Ca, V and P in the resulting sample solutions were determined by the inductively coupled plasma-optical emission spectrometer (ICP OES) Agilent 720 instrument (with standard setting).
Q6. Analysis of structure and morphology: As already reported, it is not correct to affirm that “signals in the range of 31° to 34° in the experimental patterns correspond to distinctive phosphate groups of hydroxyapatite crystal structure”, they are typical crystal planes in the calcium phosphate lattice.
Response: Thank you for pointing that out. The data has been corrected. Line -278-280
Q7. Analysis of structure and morphology: As already requested, the Authors have to report quantitative estimation about the crystallite dimensions using the Scherrer formula.
Response: The Scherrer equation is a formula that relates the size of sub-micrometre crystallites in a solid to the broadening of a peak in a diffraction pattern. It is often referred to, incorrectly, as a formula for particle size measurement or analysis. Moreover, it the case of multiphase materials it is impossible to use the Scherrer equation for particle size calculation. In our case, it was calculated by Rietveld refinement. The calculated average grain size for powders is in the range of 25 to 90 nm.
Q8. Analysis of structure and morphology: If not an aim of the present paper, the hydroxyapatite crystallinity degree is an important information and to estimate it requests 1 minute….You can use the relation reported in “F-substituted hydroxyapatite nanopowders: thermal stability, sintering behavior and mechanical properties, Ceramics International 36[1] (2010): 313-322”, properly citing this reference.
Response: The Rietveld calculation/refinement uses a least squares approach to refine a theoretical line profile until it matches the measured profile. Moreover, it can be calculated the structural parameters. The structural refinement method has several advantages over conventional quantitative analysis methods. The average grain size for the obtained materials was calculated by Rietveld method and it was in the range of 25 to 90 nm.

Reviewer 3 Report
The revised manuscript looks better but still, some questions should be revised.
Q3) It is absolutely necessary for optical materials, comparison of emission intensity is not a correct strategy (depends on numerous factors).
Q5) Please check the lines 103-104 in your revised manuscript. You proposed that this material can be used for potential bioimaging! Now, it is strange to hear that it is difficult task.
Author Response
Dear Editor,
We would like to express our sincerest gratitude to the Reviewer for the enormous efforts in criticizing the manuscript. We have considered all raised question here follows the detailed answers. Moreover, all changes we have made to the original manuscript, are marked in the red color in the text.
Comments and Suggestions for Authors:
Reviewer 3
Q1. Q3) It is absolutely necessary for optical materials, comparison of emission intensity is not a correct strategy (depends on numerous factors).
Response: It is correct that it is difficult to compare the emission intensities of different luminescence materials. Therefore, it can be used the quantum efficiency (QE) of a luminescent material. It is defined as the ratio of the number of photons emitted to the number of photons absorbed. Moreover, it could be compared the emission intensities within the same materials that are measured at the same conditions.
Q2. Q5) Please check the lines 103-104 in your revised manuscript. You proposed that this material can be used for potential bioimaging! Now, it is strange to hear that it is difficult task.
Response: Thank you for pointing that out. In the “Introduction” section was explained a potential use of these compounds as bio-imaging materials because of the lanthanide ion presence in the obtained materials. Moreover, it was also mentioned their potential use as a tissue filler. Before a research, it cannot be predicted what kind of properties the materials would be shown. It was expected that the materials doped with the Eu3+ ions could be used for the bio-imaging of the cells or living tissue. However, it has been discovered that such materials could not be used in this field. Anyway, it was confirmed that these nanosized materials showed biocompatible features and could be used in further in vivo research as a safe filler material in the soft tissue fraction. – Line 103-110

Round 3
Reviewer 1 Report
The points raised in reviewer's comments have been addressed and the manuscript has been definitively improved.
Reviewer 2 Report
The paper can be accepted in the present version. Only two specifications: 1) in the Abstract section a contextualization has not been added yet; 2) EDX analysis is used to estimate the presence of specific elements, but it is not a quantitative measurement (infact, the Authors have added ICP analysis).
Reviewer 3 Report
no more comments